# A Characterization of Semi-Supervised Adversarially Robust PAC Learnability

**Idan Attias**
Department of Computer Science
Ben-Gurion University of the Negev
idanatti@post.bgu.ac.il

**Steve Hanneke**
Department of Computer Science
Purdue University
steve.hanneke@gmail.com

**Yishay Mansour**
Blavatnik School of Computer Science
Tel Aviv University and Google Research
mansour.yishay@gmail.com

## Abstract

We study the problem of learning an adversarially robust predictor to test time attacks in the *semi-supervised* PAC model. We address the question of how many *labeled* and *unlabeled* examples are required to ensure learning. We show that having enough unlabeled data (the size of a labeled sample that a fully-supervised method would require), the labeled sample complexity can be arbitrarily smaller compared to previous works, and is sharply characterized by a *different* complexity measure. We prove nearly matching upper and lower bounds on this sample complexity. This shows that there is a significant benefit in semi-supervised robust learning even in the worst-case distribution-free model, and establishes a gap between supervised and semi-supervised label complexities which is known not to hold in standard non-robust PAC learning.

## 1 Introduction

The problem of learning predictors that are immune to adversarial corruptions at inference time is central in modern machine learning. The phenomenon of fooling learning models by adding imperceptible perturbations to their input illustrates a basic vulnerability of learning-based models and is named *adversarial examples*. We study the model of adversarially-robust PAC learning, in a *semi-supervised* setting.

Adversarial robustness has been shown to significantly benefit from semi-supervised learning, mostly empirically, but also theoretically in some specific cases of distributions [e.g., 18, 58, 51, 46, 1, 55, 36]. In this paper, we ask the following natural question. To what extent can we benefit from *unlabeled* data in the learning process of robust models in the general case? More specifically, what is the sample complexity in a distribution-free model?

Our semi-supervised model is formalized as follows. Let $\mathcal{H} \subseteq \{0,1\}^{\mathcal{X}}$ be a hypothesis class. We formalize the adversarial attack by a perturbation function $\mathcal{U} : \mathcal{X} \to 2^{\mathcal{X}}$, where $\mathcal{U}(x)$ is the set of possible perturbations (attacks) on $x$. In practice, we usually consider $\mathcal{U}(x)$ to be the $\ell_p$ ball centered at $x$. In this paper, we have no restriction on $\mathcal{U}$, besides $x \in \mathcal{U}(x)$. The robust error of hypothesis $h$ on a pair $(x, y)$ is $\sup_{z \in \mathcal{U}(x)} \mathbb{I}[h(z) \neq y]$. The learner has access to both *labeled* and *unlabeled* examples drawn i.i.d. from unknown distribution $\mathcal{D}$, and the goal is to find $h \in \mathcal{H}$ with low robust error on a random point from $\mathcal{D}$. The sample complexity in semi-supervised learning has two parameters, the number of labeled examples and the number of unlabeled examples which suffice to

ensure learning. The learner would like to restrict the amount of labeled data, which is significantly more expensive to obtain than unlabeled data.

In this paper, we show a gap between supervised and semi-supervised label complexities of adversarially robust learning in a distribution-free model. The label complexity in semi-supervised may be arbitrarily smaller compared to the supervised case and is characterized by a different complexity measure. Importantly, we are not using more data, just less labeled data. The unlabeled sample size is the same as how much labeled data a fully-supervised method would require, so this is a strict improvement. This kind of gap is known not to hold in standard (non-robust) PAC learning, this is a unique property of robust learning.

**Background.** The following complexity measure $\mathrm{VC}_{\mathcal{U}}$ was introduced by Montasser et al. [40] (and denoted there by $\dim_{\mathcal{U}\times}$) as a candidate for determining the sample complexity of supervised robust learning. It was shown that indeed its finiteness is necessary, but not sufficient. This parameter is our primary object in this work, as we will show that it characterizes the labeled sample complexity of *semi-supervised* robust PAC-learning.

**Definition 1.1** ($\mathrm{VC}_{\mathcal{U}}$-**dimension**) A sequence of points $\{x_1, \ldots, x_k\}$ is $\mathcal{U}$-*shattered* by $\mathcal{H}$ if $\forall y_1, \ldots, y_k \in \{0, 1\}, \exists h \in \mathcal{H}$ such that $\forall i \in [k], \forall z \in \mathcal{U}(x_i), h(z) = y_i$. The $\mathrm{VC}_{\mathcal{U}}(\mathcal{H})$ is largest integer $k$ for which there exists a sequence $\{x_1, \ldots, x_k\}$ $\mathcal{U}$-shattered by $\mathcal{H}$.

Intuitively, this dimension relates to the shattering of the entire perturbation sets, instead of one point in the standard VC-dimension. When $\mathcal{U}(x) = \{x\}$, this parameter coincides with the standard VC. Moreover, for any hypothesis class $\mathcal{H}$, it holds that $\mathrm{VC}_{\mathcal{U}}(\mathcal{H}) \leq \mathrm{VC}(\mathcal{H})$, and the gap can be arbitrarily large. That is, there exist $\mathcal{H}_0$ such that $\mathrm{VC}_{\mathcal{U}}(\mathcal{H}_0) = 0$ and $\mathrm{VC}(\mathcal{H}_0) = \infty$ (see Proposition 3.2).

For an improved lower bound on the sample complexity, Montasser et al. [40, Theorem 10] introduced the Robust Shattering dimension, denoted by $\mathrm{RS}_{\mathcal{U}}$ (and denoted there by $\dim_{\mathcal{U}}$).

**Definition 1.2** ($\mathrm{RS}_{\mathcal{U}}$-**dimension**) A sequence $x_1, \ldots, x_k$ is said to be $\mathcal{U}$-robustly shattered by $\mathcal{F}$ if $\exists z_1^+, z_1^-, \ldots, z_k^+, z_k^-$ such that $x_i \in \mathcal{U}\left(z_i^+\right) \cap \mathcal{U}\left(z_i^-\right) \forall i \in [k]$ and $\forall y_1, \ldots, y_k \in \{+, -\}, \exists f \in \mathcal{F}$ with $f(\zeta) = y_i, \forall \zeta \in \mathcal{U}\left(z_i^{y_i}\right), \forall i \in [k]$. The $\mathcal{U}$-robust shattering dimension $\mathrm{RS}_{\mathcal{U}}(\mathcal{H})$ is defined as the maximum size of a set that is $\mathcal{U}$-robustly shattered by $\mathcal{H}$.

Specifically, the lower bound on the sample complexity is $\Omega\left(\frac{\mathrm{RS}_{\mathcal{U}}}{\epsilon} + \frac{1}{\epsilon}\log\frac{1}{\delta}\right)$ for realizable robust learning, and $\Omega\left(\frac{\mathrm{RS}_{\mathcal{U}}}{\epsilon^2} + \frac{1}{\epsilon^2}\log\frac{1}{\delta}\right)$ for agnostic robust learning. They also showed upper bounds of $\tilde{\mathcal{O}}\left(\frac{\mathrm{VC}\cdot\mathrm{VC}^*}{\epsilon} + \frac{\log\frac{1}{\delta}}{\epsilon}\right)$[1] in the realizable case and $\tilde{\mathcal{O}}\left(\frac{\mathrm{VC}\cdot\mathrm{VC}^*}{\epsilon^2} + \frac{\log\frac{1}{\delta}}{\epsilon^2}\right)$ in the agnostic case, where $\mathrm{VC}^*$ is the dual VC dimension (definitions are in Appendix A). Montasser et al. [40] showed that for any $\mathcal{H}$, $\mathrm{VC}_{\mathcal{U}}(\mathcal{H}) \leq \mathrm{RS}_{\mathcal{U}}(\mathcal{H}) \leq \mathrm{VC}(\mathcal{H})$, and there can be an arbitrary gap between them. Specifically, there exists $\mathcal{H}_0$ with $\mathrm{VC}_{\mathcal{U}}(\mathcal{H}_0) = 0$ and $\mathrm{RS}_{\mathcal{U}}(\mathcal{H}_0) = \infty$, and there exists $\mathcal{H}_1$ with $\mathrm{RS}_{\mathcal{U}}(\mathcal{H}_1) = 0$ and $\mathrm{VC}(\mathcal{H}_1) = \infty$.

**Main contributions.**

- In Section 3, we first analyze the simple case where the support of the marginal distribution on the inputs is fully known to the learner. In this case, we show a tight bound of $\Theta\left(\frac{\mathrm{VC}_{\mathcal{U}}(\mathcal{H})}{\epsilon} + \frac{\log\frac{1}{\delta}}{\epsilon}\right)$ on the labeled complexity for learning $\mathcal{H}$.

- In Section 4, we present a generic algorithm that can be applied both for the realizable and agnostic settings. We prove an upper bound and nearly matching lower bounds on the sample complexity in the realizable case. For semi-supervised robust learning, we prove a labeled sample complexity bound $\Lambda^{\mathrm{ss}}$ and compare it to the sample complexity of supervised robust learning $\Lambda^{\mathrm{s}}$. Our algorithm uses $\Lambda^{\mathrm{ss}} = \tilde{\mathcal{O}}\left(\frac{\mathrm{VC}_{\mathcal{U}}}{\epsilon} + \frac{1}{\epsilon}\log\frac{1}{\delta}\right)$ *labeled* examples and $\mathcal{O}(\Lambda^{\mathrm{s}})$ *unlabeled* examples. Recall that $\Lambda^{\mathrm{s}} = \Omega(\mathrm{RS}_{\mathcal{U}})$, and since $\mathrm{RS}_{\mathcal{U}}$ can be arbitrarily larger than $\mathrm{VC}_{\mathcal{U}}$, this means our labeled sample complexity represents a strong improvement over the sample complexity of supervised learning.

---

[1]$\tilde{\mathcal{O}}(\cdot)$ stands for omitting poly-logarithmic factors of $\mathrm{VC}, \mathrm{VC}^*, \mathrm{VC}_{\mathcal{U}}, \mathrm{RS}_{\mathcal{U}}, 1/\epsilon, 1/\delta$.

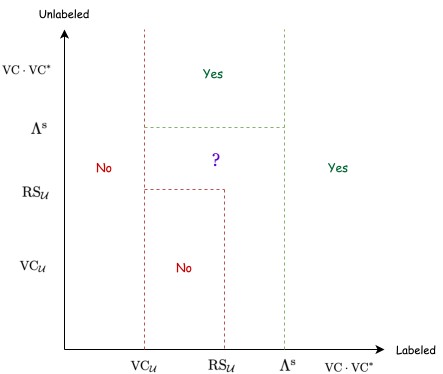

**Sample complexity for semi-supervised adversarially-robust learning**

Figure 1: Summary of the sample complexity regimes for semi-supervised robust learning, for the realizable model and the agnostic model with error $3\eta + \epsilon$, where $\eta$ is the minimal agnostic error in the hypothesis class.

Obtaining an error of $\eta + \epsilon$ requires at least $\mathrm{RS}_{\mathcal{U}}$ labeled examples, as in the supervised case.

$\Lambda^{\mathrm{s}}$ denotes the sample complexity of supervised robust learning. It is an open question whether $\Lambda^{\mathrm{s}}$ equals $\mathrm{RS}_{\mathcal{U}}$.

- In Section 5, we prove upper and lower bounds on the sample complexity in the agnostic setting. We reveal an interesting structure, which is inherently different than the realizable case. Let $\eta$ be the minimal agnostic error. If we allow an error of $3\eta + \epsilon$, it is sufficient for our algorithm to have $\Lambda^{\mathrm{ss}} = \tilde{\mathcal{O}}\left(\frac{\mathrm{VC}_{\mathcal{U}}}{\epsilon^2} + \frac{\log\frac{1}{\delta}}{\epsilon^2}\right)$ *labeled* examples and $\mathcal{O}(\Lambda^{\mathrm{s}})$ *unlabeled* examples (as in the realizable case). If we insist on having error $\eta + \epsilon$, then there is a lower bound of $\Lambda^{\mathrm{ss}} = \Omega\left(\frac{\mathrm{RS}_{\mathcal{U}}}{\epsilon^2} + \frac{1}{\epsilon^2}\log\frac{1}{\delta}\right)$ labeled examples. Furthermore, an error of $(\frac{3}{2} - \gamma)\eta + \epsilon$ is unavoidable if the learner is restricted to $\mathcal{O}(\mathrm{VC}_{\mathcal{U}})$ labeled examples, for any $\gamma > 0$. We also show that *improper* learning is necessary, similar to the supervised case. We summarize the results in Fig. 1 showing for which labeled and unlabeled samples we have a robust learner.

- The above results show that there is a significant benefit in semi-supervised robust learning. For example, take $\mathcal{H}_0$ with $\mathrm{VC}_{\mathcal{U}}(\mathcal{H}_0) = 0$ and $\mathrm{RS}_{\mathcal{U}}(\mathcal{H}_0) = n$. The labeled sample size for learning $\mathcal{H}_0$ in supervised learning is $\Omega(n)$. In contrast, in semi-supervised learning our algorithms requires only $\mathcal{O}(1)$ *labeled* examples and $\mathcal{O}(n)$ *unlabeled* examples. We are not using more data, just less labeled data. Note that $n$ can be arbitrarily large.

- A byproduct of our result is that if we assume that the distribution is robustly realizable by a hypothesis class (i.e., there exists a hypothesis with zero robust error) then, with respect to the non-robust loss (i.e., the standard 0-1 loss) we can learn with only $\tilde{\mathcal{O}}\left(\frac{\mathrm{VC}_{\mathcal{U}}(\mathcal{H})}{\epsilon} + \frac{\log\frac{1}{\delta}}{\epsilon}\right)$ labeled examples, even if the VC is infinite. Recall that there exists $\mathcal{H}_0$ with $\mathrm{VC}_{\mathcal{U}}(\mathcal{H}_0) = 0$, $\mathrm{RS}_{\mathcal{U}}(\mathcal{H}_0) = \infty$ and $\mathrm{VC}(\mathcal{H}_0) = \infty$. Learning linear functions with margin is a special case of this data-dependent assumption. Moreover, we show that this is obtained only by *improper* learning. (See Section 6.)

**Related work.** *Adversarially robust learning.* The work of Montasser et al. [40] studied the setting of fully-supervised robust PAC learning. In this paper, we propose a semi-supervised method with a significant improvement in the labeled sample size. We show that the labeled and unlabeled sample complexities are controlled by different complexity measures. Adversarially robust learning has been extensively studied in several supervised learning models [e.g., 25, 49, 34, 57, 20, 7, 35, 6, 10, 44, 41–43, 3, 11, 21, 9, 15, 56, 4]. For semi-supervised robust learning, Ashtiani et al. [3] showed that under some assumptions, robust PAC learning is possible with $\mathcal{O}(\mathrm{VC}(\mathcal{H}))$ labeled examples and additional unlabeled samples. Carmon et al. [18] studied a robust semi-supervised setting where the distribution is a mixture of Gaussians and the hypothesis class is linear separators.

*Semi-supervised (non-robust) learning.* There is substantial interest in semi-supervised (non-robust) learning, and many contemporary practical problems significantly benefit from it [e.g., 16, 19, 59]. This was formalized in theoretical frameworks. Urner et al. [52] suggested a semi-supervised learning (non-robust) framework, with an algorithmic idea that is similar to our method. Their framework consists of two steps; using labeled data to learn a classifier with a small error (not necessarily a member of the target class $\mathcal{H}$), and then labeling an unlabeled input sample in order to use a fully-supervised proper learner. They investigate scenarios where the saving of labeled examples occurs. In our paper, we are interested in the robust loss function. We use labeled data in order to learn a classifier (with the 0-1 loss function) from a class with a potentially smaller complexity

measure, then we label an unlabeled input sample, and use a fully-supervised method using the robust loss function. The sample complexity of learning the robust loss class is controlled by a larger complexity measure. Fortunately, this affects our unlabeled sample size and not the labeled sample size as in the fully-supervised setting. Göpfert et al. [27] studied circumstances where the learning rate can be improved given unlabeled data. Darnstädt et al. [23] showed that the label complexity gap between the semi-supervised and the fully supervised setting can become arbitrarily large for concept classes of infinite VC-dimension, and this gap is bounded when a function class contains the constant zero and the constant one functions. Balcan and Blum [13, 12] introduced an augmented version of the PAC model designed for semi-supervised learning and analyzed when unlabeled data can help. The main idea is to augment the notion of learning a concept class, with a notion of compatibility between a function and the data distribution that we hope the target function will satisfy.

## 2 Preliminaries

Let $\mathcal{X}$ be the instance space, $\mathcal{Y}$ a label space, and $\mathcal{H} \subseteq \mathcal{Y}^{\mathcal{X}}$ a hypothesis class. A perturbation function $\mathcal{U} : \mathcal{X} \rightarrow 2^{\mathcal{X}}$ maps an input to a set $\mathcal{U}(x) \subseteq \mathcal{X}$. Denote the 0-1 loss of hypothesis $h$ on $(x, y)$ by $\ell_{\text{0-1}}(h; x, y) = \mathbb{I}[h(x) \neq y]$, and the robust loss with respect to $\mathcal{U}$ by $\ell_{\mathcal{U}}(h; x, y) = \sup_{z \in \mathcal{U}(x)} \mathbb{I}[h(z) \neq y]$. Denote the support of a distribution $\mathcal{D}$ over $\mathcal{X} \times \mathcal{Y}$ by $\text{supp}(\mathcal{D}) = \{(x, y) \in \mathcal{X} \times \mathcal{Y} : \mathcal{D}(x, y) > 0\}$. Denote the marginal distribution $\mathcal{D}_{\mathcal{X}}$ on $\mathcal{X}$ and its support by $\text{supp}(\mathcal{D}_{\mathcal{X}}) = \{x \in \mathcal{X} : \mathcal{D}(x, y) > 0\}$. Define the *robust risk* of a hypothesis $h \in \mathcal{H}$ with respect to distribution $\mathcal{D}$ over $\mathcal{X} \times \mathcal{Y}$,

$$\text{R}_{\mathcal{U}}(h; \mathcal{D}) = \mathbb{E}_{(x,y) \sim \mathcal{D}}[\ell_{\mathcal{U}}(h; x, y)] = \mathbb{E}_{(x,y) \sim \mathcal{D}}\left[\sup_{z \in \mathcal{U}(x)} \mathbb{I}[h(z) \neq y]\right].$$

The approximation error of $\mathcal{H}$ on $\mathcal{D}$, namely, the optimal robust error achievable by a hypothesis in $\mathcal{H}$ on $\mathcal{D}$ is denoted by,

$$\text{R}_{\mathcal{U}}(\mathcal{H}; \mathcal{D}) = \inf_{h \in \mathcal{H}} \text{R}_{\mathcal{U}}(h; \mathcal{D}).$$

We say that a distribution $\mathcal{D}$ is *robustly realizable* by a class $\mathcal{H}$ if $\text{R}_{\mathcal{U}}(\mathcal{H}; \mathcal{D}) = 0$.

Define the *empirical robust risk* of a hypothesis $h \in \mathcal{H}$ with respect to a sequence $S \in (\mathcal{X} \times \mathcal{Y})^*$,

$$\widehat{\text{R}}_{\mathcal{U}}(h; S) = \frac{1}{|S|} \sum_{(x,y) \in S} \ell_{\mathcal{U}}(h; x, y) = \frac{1}{|S|} \sum_{(x,y) \in S} \left[\sup_{z \in \mathcal{U}(x)} \mathbb{I}[h(z) \neq y]\right].$$

The *robust empirical risk minimizer* learning algorithm $\text{RERM} : (\mathcal{X} \times \mathcal{Y})^* \rightarrow \mathcal{H}$ for a class $\mathcal{H}$ on a sequence $S$ is defined by

$$\text{RERM}_{\mathcal{H}}(S) \in \underset{h \in \mathcal{H}}{\text{argmin}}\ \widehat{\text{R}}_{\mathcal{U}}(h; S).$$

When the perturbation function is the identity, $\mathcal{U}(x) = \{x\}$, we recover the standard notions. The *risk* of a hypothesis $h \in \mathcal{H}$ with respect to distribution $\mathcal{D}$ over $\mathcal{X} \times \mathcal{Y}$ is defined by $\text{R}(h; \mathcal{D}) = \mathbb{E}_{(x,y) \sim \mathcal{D}}[\ell_{\text{0-1}}(h; x, y)] = \mathbb{E}_{(x,y) \sim \mathcal{D}}[\mathbb{I}[h(x) \neq y]]$, and the *empirical risk* of a hypothesis $h \in \mathcal{H}$ with respect to a sequence $S \in (\mathcal{X} \times \mathcal{Y})^*$ is defined by $\widehat{\text{R}}(h; S) = \frac{1}{|S|} \sum_{(x,y) \in S} \ell_{\text{0-1}}(h; x, y) = \frac{1}{|S|} \sum_{(x,y) \in S} [\mathbb{I}[h(x) \neq y]]$. The *empirical risk minimizer* learning algorithm $\text{ERM} : (\mathcal{X} \times \mathcal{Y})^* \rightarrow \mathcal{H}$ for a class $\mathcal{H}$ on a sequence $S$ is defined by $\text{ERM}_{\mathcal{H}}(S) \in \text{argmin}_{h \in \mathcal{H}}\ \widehat{\text{R}}(h; S)$.

A learning algorithm $\mathcal{A} : (\mathcal{X} \times \mathcal{Y})^* \rightarrow \mathcal{Y}^{\mathcal{X}}$ for a class $\mathcal{H}$ is called *proper* if it always outputs a hypothesis in $\mathcal{H}$, otherwise it is called *improper*.

**Realizable robust** PAC **learning.** We define the supervised and semi-supervised settings.

**Definition 2.1 (Realizable robust** PAC **learnability)** For any $\epsilon, \delta \in (0, 1)$, the sample complexity of realizable robust $(\epsilon, \delta)$-PAC learning for a class $\mathcal{H}$, with respect to perturbation function $\mathcal{U}$, denoted by $\Lambda_{\text{RE}}(\epsilon, \delta, \mathcal{H}, \mathcal{U})$, is the smallest integer $m$ for which there exists a learning algorithm $\mathcal{A} : (\mathcal{X} \times \mathcal{Y})^* \rightarrow \mathcal{Y}^{\mathcal{X}}$, such that for every distribution $\mathcal{D}$ over $\mathcal{X} \times \mathcal{Y}$ robustly realizable by $\mathcal{H}$, namely $\text{R}_{\mathcal{U}}(\mathcal{H}; D) = 0$, for a random sample $S \sim \mathcal{D}^m$, it holds that

$$\mathbb{P}(\text{R}_{\mathcal{U}}(\mathcal{A}(S); D) \leq \epsilon) > 1 - \delta.$$

If no such $m$ exists, define $\Lambda_{\mathrm{RE}}(\epsilon, \delta, \mathcal{H}, \mathcal{U}) = \infty$, and $\mathcal{H}$ is not robustly $(\epsilon, \delta)$-PAC learnable with respect to $\mathcal{U}$.

For the standard (non-robust) learning with the 0-1 loss function, we omit the dependence on $\mathcal{U}$ and denote the sample complexity of class $\mathcal{H}$ by $\Lambda_{\mathrm{RE}}(\epsilon, \delta, \mathcal{H})$.

**Definition 2.2 (Realizable semi-supervised robust** PAC **learnability)** A hypothesis class $\mathcal{H}$ is semi-supervised realizable robust $(\epsilon, \delta)$-PAC learnable, with respect to perturbation function $\mathcal{U}$, if for any $\epsilon, \delta \in (0, 1)$, there exists $m_u, m_l \in \mathbb{N} \cup \{0\}$, and a learning algorithm $\mathcal{A} : (\mathcal{X} \times \mathcal{Y})^* \cup (\mathcal{X})^* \to \mathcal{Y}^{\mathcal{X}}$, such that for every distribution $\mathcal{D}$ over $\mathcal{X} \times \mathcal{Y}$ robustly realizable by $\mathcal{H}$, namely $\mathrm{R}_{\mathcal{U}}(\mathcal{H}; D) = 0$, for random samples $S^l \sim \mathcal{D}^{m_l}$ and $S^u_{\mathcal{X}} \sim \mathcal{D}_{\mathcal{X}}^{m_u}$, it holds that

$$\mathbb{P}\left(\mathrm{R}_{\mathcal{U}}\left(\mathcal{A}(S^l, S^u_{\mathcal{X}}); D\right) \le \epsilon\right) > 1 - \delta.$$

The sample complexity $\mathcal{M}_{\mathrm{RE}}(\epsilon, \delta, \mathcal{H}, \mathcal{U})$ includes all such pairs $(m_u, m_l)$. If no such $(m_u, m_l)$ exist, then $\mathcal{M}_{\mathrm{RE}}(\epsilon, \delta, \mathcal{H}, \mathcal{U}) = \emptyset$.

**Agnostic robust** PAC **learning.** In this case we have $\mathrm{R}_{\mathcal{U}}(\mathcal{H}; \mathcal{D}) > 0$, and we would like to compete with the optimal $h \in \mathcal{H}$. We add a parameter to the sample complexity, denoted by $\eta$, which is the optimal robust error of a hypothesis in $\mathcal{H}$, namely $\eta = \mathrm{R}_{\mathcal{U}}(\mathcal{H}; \mathcal{D})$. We say that a function $f$ is $(\alpha, \epsilon)$-optimal if $\mathrm{R}_{\mathcal{U}}(f; D) \le \alpha\eta + \epsilon$.

**Definition 2.3 (Agnostic robust** PAC **learnability)** For any $\epsilon, \delta \in (0, 1)$, the sample complexity of agnostic robust $(\alpha, \epsilon, \delta)$-PAC learning for a class $\mathcal{H}$, with respect to perturbation function $\mathcal{U}$, denoted by $\Lambda_{\mathrm{AG}}(\alpha, \epsilon, \delta, \mathcal{H}, \mathcal{U}, \eta)$, is the smallest integer $m$, for which there exists a learning algorithm $\mathcal{A} : (\mathcal{X} \times \mathcal{Y})^* \to \mathcal{Y}^{\mathcal{X}}$, such that for every distribution $\mathcal{D}$ over $\mathcal{X} \times \mathcal{Y}$, for a random sample $S \sim \mathcal{D}^m$, it holds that

$$\mathbb{P}\left(\mathrm{R}_{\mathcal{U}}\left(\mathcal{A}(S); D\right) \le \alpha \inf_{h \in \mathcal{H}} \mathrm{R}_{\mathcal{U}}(h; \mathcal{D}) + \epsilon\right) > 1 - \delta.$$

If no such $m$ exists, define $\Lambda_{\mathrm{AG}}(\alpha, \epsilon, \delta, \mathcal{H}, \mathcal{U}, \eta) = \infty$, and $\mathcal{H}$ is not robustly $(\alpha, \epsilon, \delta)$-PAC learnable in the agnostic setting with respect to $\mathcal{U}$. Note that for $\alpha = 1$ we recover the standard agnostic definition, our notation allows for a more relaxed approximation.

Analogously, we define the semi-supervised case.

**Definition 2.4 (Agnostic semi-supervised robust** PAC **learnability)** A hypothesis class $\mathcal{H}$ is semi-supervised agnostically robust $(\alpha, \epsilon, \delta)$-PAC learnable, with respect to perturbation function $\mathcal{U}$, if for any $\epsilon, \delta \in (0, 1)$, there exists $m_u, m_l \in \mathbb{N} \cup \{0\}$, and a learning algorithm $\mathcal{A} : (\mathcal{X} \times \mathcal{Y})^* \cup (\mathcal{X})^* \to \mathcal{Y}^{\mathcal{X}}$, such that for every distribution $\mathcal{D}$ over $\mathcal{X} \times \mathcal{Y}$, for random samples $S^l \sim \mathcal{D}^{m_l}$ and $S^u_{\mathcal{X}} \sim \mathcal{D}_{\mathcal{X}}^{m_u}$, it holds that

$$\mathbb{P}\left(\mathrm{R}_{\mathcal{U}}\left(\mathcal{A}(S^l, S^u_{\mathcal{X}}); \mathcal{D}\right) \le \alpha \inf_{h \in \mathcal{H}} \mathrm{R}_{\mathcal{U}}(h; \mathcal{D}) + \epsilon\right) > 1 - \delta.$$

The sample complexity $\mathcal{M}_{\mathrm{AG}}(\alpha, \epsilon, \delta, \mathcal{H}, \mathcal{U}, \eta)$ includes all such pairs $(m_u, m_l)$. If no such $(m_u, m_l)$ exist, then $\mathcal{M}_{\mathrm{AG}}(\alpha, \epsilon, \delta, \mathcal{H}, \mathcal{U}, \eta) = \emptyset$.

**Partial concept classes [2].** Let a partial concept class $\mathcal{H} \subseteq \{0, 1, \star\}^{\mathcal{X}}$. For $h \in \mathcal{H}$ and input $x$ such that $h(x) = \star$, we say that $h$ is undefined on $x$. The support of a partial hypothesis $h : \mathcal{X} \to \{0, 1, \star\}$ is the preimage of $\{0, 1\}$, formally, $h^{-1}(\{0, 1\}) = \{x \in \mathcal{X} : h(x) \ne \star\}$. The main motivation for introducing partial concept classes is that data-dependent assumptions can be modeled in a natural way that extends the classic theory of total concepts. The VC dimension of a partial class $\mathcal{H}$ is defined as the maximum size of a shattered set $S \subseteq \mathcal{X}$, where $S$ is shattered by $\mathcal{H}$ if the projection of $\mathcal{H}$ on $S$ contains all possible binary patterns, $\{0, 1\}^S \subseteq \mathcal{H}|_S$. The VC-dimension also characterizes verbatim the PAC learnability of partial concept classes, even though uniform convergence does not hold in this setting.

We use the notation $\tilde{\mathcal{O}}(\cdot)$ for omitting poly-logarithmic factors of $\text{VC}, \text{VC}^*, \text{VC}_{\mathcal{U}}, \text{RS}_{\mathcal{U}}, 1/\epsilon, 1/\delta$. See Appendix A for additional preliminaries on complexity measures, sample compression schemes, and partial concept classes.

## 3 Warm-up: knowing the support of the marginal distribution

In this section, we provide a tight bound on the labeled sample complexity when the support of marginal distribution is fully known to the learner, under the robust realizable assumption. Studying this setting gives an intuition for the general semi-supervised model. The main idea is that as long as we know the support of the marginal distribution, $\text{supp}(\mathcal{D}_{\mathcal{X}}) = \{x \in \mathcal{X} : \exists y \in \mathcal{Y}, \text{ s.t. } \mathcal{D}(x, y) > 0\}$, we can restrict our search to a subspace of functions that are robustly self-consistent, $\mathcal{H}_{\mathcal{U}\text{-cons}} \subseteq \mathcal{H}$, where

$$\mathcal{H}_{\mathcal{U}\text{-cons}} = \{h \in \mathcal{H} : \forall x \in \text{supp}(\mathcal{D}_{\mathcal{X}}), \forall z, z' \in \mathcal{U}(x), h(z) = h(z')\} .$$

As long as the distribution is robustly realizable, i.e., $\text{R}_{\mathcal{U}}(\mathcal{H}; \mathcal{D}) = 0$, we are guaranteed that the target hypothesis belongs to $\mathcal{H}_{\mathcal{U}\text{-cons}}$. As a result, it suffices to learn the class $\mathcal{H}_{\mathcal{U}\text{-cons}}$ with the 0-1 loss function, in order to robustly learn the original class $\mathcal{H}$. We observe that,

$$\text{VC}(\mathcal{H}_{\mathcal{U}\text{-cons}}) = \text{VC}_{\mathcal{U}}(\mathcal{H}) \leq \text{VC}(\mathcal{H}).$$

Moreover, there exists $\mathcal{H}_0$ with $\text{VC}_{\mathcal{U}}(\mathcal{H}_0) = 0$ and $\text{VC}(\mathcal{H}_0) = \infty$ (see Proposition 3.2). Fortunately, moving from $\text{VC}(\mathcal{H})$ to $\text{VC}_{\mathcal{U}}(\mathcal{H})$ implies a significant sample complexity improvement. Since $\text{supp}(\mathcal{D}_{\mathcal{X}})$ is known, we can now employ any algorithm for learning the hypothesis class $\mathcal{H}_{\mathcal{U}\text{-cons}}$. [2] This leads eventually to robustly learning $\mathcal{H}$ with labeled sample complexity that scales linearly with $\text{VC}_{\mathcal{U}}$ (instead of the VC). Formally,

**Theorem 3.1** *For hypothesis class $\mathcal{H}$ and adversary $\mathcal{U}$, when the support of the marginal distribution $\mathcal{D}_{\mathcal{X}}$ is known, the labeled sample complexity is $\Theta\left(\frac{\text{VC}_{\mathcal{U}}(\mathcal{H})}{\epsilon} + \frac{\log \frac{1}{\delta}}{\epsilon}\right)$.*

**Remark.** The statement in the theorem is about the labeled sample size that is required for learning when the marginal distribution is known. This is different than the sample complexity in Definition 2.2, where we ask about the labeled and unlabeled sample sizes required for learning. Here we make a strong assumption of knowing the marginal distribution, instead of having access to an unlabeled sample.

The following Proposition demonstrates that semi-supervised robust learning requires much fewer labeled samples compared to the supervised counterpart. Recall the lower bound on the sample complexity of supervised robust learning, $\Lambda_{\text{RE}}(\epsilon, \delta, \mathcal{H}, \mathcal{U}) = \Omega\left(\frac{\text{RS}_{\mathcal{U}}(\mathcal{H})}{\epsilon} + \frac{1}{\epsilon}\log \frac{1}{\delta}\right)$ given by Montasser et al. [40, Theorem 10]. For completeness, we prove the following in Appendix B.

**Proposition 3.2 ([40], Proposition 9)** *There exists a hypothesis class $\mathcal{H}_0$ such that $\text{VC}_{\mathcal{U}}(\mathcal{H}_0) = 0$, $\text{RS}_{\mathcal{U}}(\mathcal{H}_0) = \infty$, and $\text{VC}(\mathcal{H}_0) = \infty$.*

We can now conclude the following separation result on supervised and semi-supervised label complexities.

**Corollary 3.3** *The hypothesis class in Proposition 3.2 is not learnable in supervised robust learning (i.e., we need to see the entire data distribution). However, when $\text{supp}(\mathcal{D}_{\mathcal{X}})$ is known, this class can be learned with $\mathcal{O}(\frac{1}{\epsilon}\log\frac{1}{\delta})$ labeled examples.*

In the next section, we prove a stronger separation in the general semi-supervised setting. The size of the labeled data required in the supervised case is lower bounded by $\text{RS}_{\mathcal{U}}$, whereas in the semi-supervised case, the *labeled* sample complexity depends only on $\text{VC}_{\mathcal{U}}$ and the *unlabeled* data is

---

[2]See Mohri et al. [39, Chapter 3] for standard upper and lower bounds. In order to remove the superfluous $\log \frac{1}{\epsilon}$ factor of the standard uniform convergence based upper bound, $\mathcal{O}\left(\frac{\text{VC}_{\mathcal{U}}(\mathcal{H})}{\epsilon}\log\frac{1}{\epsilon} + \frac{\log\frac{1}{\delta}}{\epsilon}\right)$, we can use the learning algorithm and its analysis from Hanneke [30] that applies for any $\mathcal{H}$ and $\mathcal{D}$, or some other algorithms that are doing so while restricting the hypothesis class or the data distribution [e.g., 8, 22, 31, 29, 37, 26, 17, 14].

lower bounded by $\mathrm{RS}_{\mathcal{U}}$. Moreover, note that in Theorem 3.1, when $\mathrm{supp}(\mathcal{D}_{\mathcal{X}})$ is known, we can use any proper learner. In Section 4 we show that in the general semi-supervised model this is not the case, and sometimes improper learning is necessary, similar to supervised robust learning.

## 4   Near-optimal semi-supervised sample complexity

In this section, we present our algorithm and its guarantees for the realizable setting. We also prove nearly matching lower bounds on the sample complexity. Finally, we show that improper learning is necessary in semi-supervised robust learning, similar to the supervised case.

We present a generic semi-supervised robust learner, that can be applied in both realizable and agnostic settings. The algorithm uses the following two subroutines. The first one is any algorithm for learning partial concept classes, which controls our *labeled* sample size. (In Appendix F we discuss in detail the algorithm suggested by Alon et al. [2].) The second subroutine is any algorithm for the agnostic adversarially robust supervised learning, which controls our *unlabeled* sample size. (In Appendix G we discuss in detail the algorithm suggested by Montasser et al. [40].) Any progress on one of these problems improves directly the guarantees of our algorithm. We use the following definition that explains how to convert a total concept class into a partial one, in a way that preserves the idea of the robust loss function.

**Definition 4.1** Let a hypothesis class $\mathcal{H} \subseteq \{0,1\}^{\mathcal{X}}$ and a perturbation function $\mathcal{U} : \mathcal{X} \to 2^{\mathcal{X}}$. For any $h \in \mathcal{H}$, we define a corresponding partial concept $h^{\star} : \mathcal{X} \to \{0,1,\star\}$, and denote this mapping by $\varphi(h) = h^{\star}$. For $x \in \mathcal{X}$, whenever $h$ is not consistent on the entire set $\mathcal{U}(x)$, i.e., $\exists z, z' \in \mathcal{U}(x), h(z) \neq h(z')$, define $h^{\star}(x) = \star$. Otherwise, $h$ is robustly self-consistent on $x$, i.e., $\forall z, z' \in \mathcal{U}(x), h(z) = h(z')$ and $h$ remains unchanged, $h^{\star}(x) = h(x)$. The corresponding partial concept class is defined by $\mathcal{H}_{\mathcal{U}}^{\star} = \{h^{\star} : \varphi(h) = h^{\star}, \ \forall h \in \mathcal{H}\}$.

The main motivation for the above definition is the following. Fix a hypothesis $h$. For any point $x$, as defined above, the adversary can force a mistake on $h$, regardless of the prediction of $h$. We would like to mark such points as *mistake*. We do this by defining a partial concept $h^{\star}$ and setting $h^{\star}(x) = \star$, which, for partial concepts, implies a mistake. The benefit of this preprocessing is that we reduce the complexity of the hypothesis class from VC to $\mathrm{VC}_{\mathcal{U}}$, which potentially can reduce the labeled sample complexity. We are now ready to describe the algorithm.

---

**Algorithm 1** Generic Adversarially-Robust Semi-Supervised (GRASS) learner

---

**Input:** Labeled data set $S^l \sim \mathcal{D}^{m_l}$, unlabeled data set $S^u_{\mathcal{X}} \sim \mathcal{D}^{m_u}_{\mathcal{X}}$, hypothesis class $\mathcal{H}$, perturbation function $\mathcal{U}$, parameters $\epsilon, \delta$.
**Algorithms used:** PAC learner $\mathcal{A}$ for partial concept classes, agnostic adversarially robust supervised PAC learner $\mathcal{B}$.

1. Given the class $\mathcal{H}$, construct the hypothesis class $\mathcal{H}_{\mathcal{U}}^{\star}$ using Definition 4.1.

2. Execute the learning algorithm for partial concepts $\mathcal{A}$ on $\mathcal{H}_{\mathcal{U}}^{\star}$ and sample $S^l$, with the 0-1 loss and parameters $\frac{\epsilon}{3}, \frac{\delta}{2}$. Denote the resulting hypothesis $h_1$.

3. Label the unlabeled data set $S^u_{\mathcal{X}}$ with $h_1$, denote the labeled sample by $S^u$. (On points where $h_1$ predicts $\star$, we can arbitrarily choose a label of 0 or 1.)

4. Execute the agnostic adversarially robust supervised PAC learner $\mathcal{B}$ on $S^u$ with parameters $\frac{\epsilon}{3}, \frac{\delta}{2}$. Denote the resulting hypothesis $h_2$.

**Output:** $h_2$.

---

**Algorithm motivation.**   The main idea behind the algorithm is the following. Given the class $\mathcal{H}_{\mathcal{U}}^{\star}$, we would like to find a hypothesis $h_1 \in \mathcal{H}_{\mathcal{U}}^{\star}$ which has a small error, whose existence follows from our realizability assumption. The required sample size scales with $\mathrm{VC}_{\mathcal{U}}$, which is the complexity of $\mathcal{H}_{\mathcal{U}}^{\star}$, rather than VC. This is where we make a significant gain in the labeled sample complexity. Note that $h_1$ does not guarantee a small robust error, although it does guarantee a small non-robust error. We utilize an additional unlabeled sample for this task, which we label using $h_1$. If we would simply minimize the non-robust error on this sample we would simply get back $h_1$. The main insight

is that we would like to minimize the robust error over this sample, which will result in hypothesis $h_2$. We now need to bound the robust error of $h_2$. The optimal function $h_{\text{opt}}$ has only a slightly increased robust error on this sample, namely, at most on the sample points where it disagrees with $h_1$. Note that $h_1$ might have a large robust error due to the perturbation $\mathcal{U}$. However, a robust supervised PAC learner would return a hypothesis $h_2$ which has robust error similar to $h_{\text{opt}}$, which is at most $\epsilon$.

**Algorithm outline and guarantees.** In the first step, we convert $\mathcal{H}$ to $\mathcal{H}_{\mathcal{U}}^{\star}$. Then we employ a learning algorithm $\mathcal{A}$ for partial concepts on $\mathcal{H}_{\mathcal{U}}^{\star}$ with a labeled sample $S^l \sim \mathcal{D}^{m_l}$. The output of the algorithm is a function $h_1$ with $\epsilon/3$ on the 0-1 error. Crucially, we needed for this step $|S^l| = \tilde{\mathcal{O}}(\text{VC}_{\mathcal{U}}(\mathcal{H})/\epsilon)$ labeled examples for learning the partial concept $\mathcal{H}_{\mathcal{U}}^{\star}$, since $\text{VC}(\mathcal{H}_{\mathcal{U}}^{\star}) = \text{VC}_{\mathcal{U}}(\mathcal{H})$. So our labeled sample size is controlled by the sample complexity for learning partial concepts with the 0-1 loss. In step 3, we label an independent unlabeled sample $S_{\mathcal{X}}^u \sim \mathcal{D}_{\mathcal{X}}^{m_u}$ with $h_1$, denote his labeled sample by $S^u$. Define a distribution $\tilde{\mathcal{D}}$ over $\mathcal{X} \times \mathcal{Y}$ by $\tilde{\mathcal{D}}(x, h_1(x)) = \mathcal{D}_{\mathcal{X}}(x)$, and so $S^u$ is an i.i.d. sample from $\tilde{\mathcal{D}}$. We argue that the robust error of $\mathcal{H}$ with respect to $\tilde{\mathcal{D}}$ is at most $\frac{\epsilon}{3}$, i.e., $\text{R}_{\mathcal{U}}(\mathcal{H}; \tilde{\mathcal{D}}) = \frac{\epsilon}{3}$. Indeed, the function with zero robust error on $\mathcal{D}$, $h_{\text{opt}} \in \text{argmin}_{h \in \mathcal{H}} \text{R}_{\mathcal{U}}(h; \mathcal{D})$ has a robust error of at most $\frac{\epsilon}{3}$ on $\tilde{\mathcal{D}}$. Finally, we employ an agnostic adversarially robust supervised PAC learner $\mathcal{B}$ for the class $\mathcal{H}$ on $S^u \sim \tilde{\mathcal{D}}^{m_u}$, that should be of a size of the sample complexity of agnostically robust learn $\mathcal{H}$ with respect to $\mathcal{U}$, when the optimal robust error of hypothesis from $\mathcal{H}$ on $\tilde{\mathcal{D}}$ is at most $\frac{\epsilon}{3}$. Moreover, the total variation distance between $\mathcal{D}$ and $\tilde{\mathcal{D}}$ is at most $\frac{\epsilon}{3}$. We are guaranteed that the resulting hypothesis $h_2$ has a robust error of at most $\frac{\epsilon}{3} + \frac{\epsilon}{3} + \frac{\epsilon}{3} = \epsilon$ on $\mathcal{D}$. We conclude that a size of $|S_{\mathcal{X}}^u| = m_u = \Lambda_{\text{AG}}\left(1, \frac{\epsilon}{3}, \frac{\delta}{2}, \mathcal{H}, \mathcal{U}, \eta = \frac{\epsilon}{3}\right)$ unlabeled samples suffices, this completes the proof for Theorem 4.2. For a specific instantiation of such an algorithm ([40]), we deduce the sample complexity in Theorem 4.4. A simple analysis of the latter yields a dependence of $\epsilon^2$ for the unlabeled sample size. However, by applying a suitable data-dependent generalization bound, we reduce this dependence to $\epsilon$. (Full proofs appear in Appendix C).

We now formally present the sample complexity of the generic semi-supervised learner for the robust realizable setting. First, in the case of using a generic agnostic robust supervised learner as a subroutine (step 4 in the algorithm). Then we deduce the sample complexity of a specific instantiation of such an algorithm.

**Theorem 4.2** *For any hypothesis class $\mathcal{H}$ and adversary $\mathcal{U}$, algorithm* GRASS $(\epsilon, \delta)$-PAC *learns $\mathcal{H}$ with respect to the robust loss function, in the realizable robust case, with samples of size*

$$m_l = \mathcal{O}\left(\frac{\text{VC}_{\mathcal{U}}(\mathcal{H})}{\epsilon}\log^2\frac{\text{VC}_{\mathcal{U}}(\mathcal{H})}{\epsilon} + \frac{\log\frac{1}{\delta}}{\epsilon}\right) \;,\; m_u = \Lambda_{\text{AG}}\left(1, \frac{\epsilon}{3}, \frac{\delta}{2}, \mathcal{H}, \mathcal{U}, \eta = \frac{\epsilon}{3}\right),$$

*where $\Lambda_{\text{AG}}(\alpha, \epsilon, \delta, \mathcal{H}, \mathcal{U}, \eta)$ is the sample complexity of adversarially-robust agnostic supervised $(\alpha, \epsilon, \delta)$-PAC learning, such that $\eta$ is the error of the optimal hypothesis in $\mathcal{H}$, i.e., $\eta = \text{R}_{\mathcal{U}}(\mathcal{H}; \mathcal{D})$.*

**Remark 4.3** Note that if we simply invoke a PAC learner (for total concept classes) on $\mathcal{H}$ with the 0-1 loss, instead of steps 1 and 2 in the algorithm, we would get a labeled sample complexity of roughly $\mathcal{O}(\text{VC}(\mathcal{H}))$. This is already an exponential improvement upon previous results that require roughly $\mathcal{O}(2^{\text{VC}(\mathcal{H})})$ labeled samples. The purpose of using partial concept classes is to further reduce the labeled sample complexity to $\mathcal{O}(\text{VC}_{\mathcal{U}}(\mathcal{H}))$.

The following result follows by using the agnostic supervised robust learner suggested by Montasser et al. [40]. A simple analysis of the latter yields a dependence of $\epsilon^2$ for the unlabeled sample size. However, by applying a suitable data-dependent generalization bound, we reduce this dependence to $\epsilon$.

**Theorem 4.4** *For any hypothesis class $\mathcal{H}$ and adversary $\mathcal{U}$, Algorithm* GRASS $(\epsilon, \delta)$-PAC *learns $\mathcal{H}$ with respect to the robust loss function, in the realizable robust case, with samples of size*

$$m_l = \mathcal{O}\left(\frac{\text{VC}_{\mathcal{U}}(\mathcal{H})}{\epsilon}\log^2\frac{\text{VC}_{\mathcal{U}}(\mathcal{H})}{\epsilon} + \frac{\log\frac{1}{\delta}}{\epsilon}\right) \;,\; m_u = \tilde{\mathcal{O}}\left(\frac{\text{VC}(\mathcal{H})\,\text{VC}^*(\mathcal{H})}{\epsilon} + \frac{\log\frac{1}{\delta}}{\epsilon}\right).$$

We present nearly matching lower bounds for the realizable setting. The following Corollary stems from Theorem 3.1 and Montasser et al. [40, Theorem 10].

**Corollary 4.5** *For any $\epsilon, \delta \in (0, 1)$, the sample complexity of realizable robust $(\epsilon, \delta)$-PAC learning for a class $\mathcal{H}$, with respect to perturbation function $\mathcal{U}$ is*

$$m_l = \Omega\left(\frac{\mathrm{VC}_{\mathcal{U}}(\mathcal{H})}{\epsilon} + \frac{\log\frac{1}{\delta}}{\epsilon}\right) \,, \; m_u = \infty, \quad or \quad m_l + m_u = \Omega\left(\frac{\mathrm{RS}_{\mathcal{U}}(\mathcal{H})}{\epsilon} + \frac{\log\frac{1}{\delta}}{\epsilon}\right).$$

**Proper vs. improper.** In Section 3, we have seen that when the support of the marginal distribution $\mathcal{D}_{\mathcal{X}}$ is known, the labeled sample complexity is $\Theta\left(\frac{\mathrm{VC}_{\mathcal{U}}(\mathcal{H})}{\epsilon} + \frac{\log\frac{1}{\delta}}{\epsilon}\right)$. This was obtained by a proper learner: keep the robustly self-consistent hypotheses, $\mathcal{H}_{\mathcal{U}\text{-cons}} \subseteq \mathcal{H}$, and then use ERM on this class. The case when $\mathcal{D}_{\mathcal{X}}$ is unknown is different. We know that there exists a perturbation function $\mathcal{U}$ and a hypothesis class $\mathcal{H}$ with finite VC-dimension that is not robustly PAC learnable by any proper learning rule [40, Lemma 3]. The same proof holds in the semi-supervised case. Note that both algorithms $\mathcal{A}$ and $\mathcal{B}$ used in Algorithm 1 are improper. (The proof appears in Appendix C.)

**Theorem 4.6** *There exists $\mathcal{H}$ with $\mathrm{VC}(\mathcal{H}) = 0$ such that for any proper learning rule $\mathcal{A} : (\mathcal{X} \times \mathcal{Y})^* \cup (\mathcal{X})^* \to \mathcal{H}$, there exists a distribution $\mathcal{D}$ over $\mathcal{X} \times \mathcal{Y}$ that is robustly realizable by $\mathcal{H}$, i.e., $\mathrm{R}_{\mathcal{U}}(\mathcal{H}; \mathcal{D}) = 0$. It holds that $\mathrm{R}_{\mathcal{U}}\left(\mathcal{A}(S^l, S^u_{\mathcal{X}}); D\right) > \frac{1}{8}$ with probability at least $\frac{1}{7}$ over $S^l \sim \mathcal{D}^{m_l}$ and $S^u_{\mathcal{X}} \sim \mathcal{D}^{m_u}$, where $m_l, m_u \in \mathbb{N} \cup \{0\}$ is the size of the labeled and unlabeled samples respectively. Moreover, when the support of the marginal distribution $\mathcal{D}_{\mathcal{X}}$ is known, there exists a proper learning rule for any $\mathcal{H}$.*

# 5 Agnostic robust learning

In this section, we prove the guarantees of Algorithm 1 in the more challenging agnostic robust setting. We then prove lower bounds on the sample complexity which exhibit that it is inherently different from the realizable case.

We follow the same steps as in the proof of the realizable case, with the following important difference. In the first two steps of the algorithm, we learn a partial concept class with respect to the 0-1 loss and obtain a hypothesis with an error of $\eta + \epsilon/3$ ($\eta$ is the optimal robust error of a hypothesis in $\mathcal{H}$ and not 0). This leads eventually to an error of $3\eta + \epsilon$ for learning with respect to the robust loss.

We then present two negative results. In Theorem 5.2 we show that for obtaining error $\eta + \epsilon$ there is a lower bound of $\Omega(\mathrm{RS}_{\mathcal{U}})$ labeled examples, this result coincides with the lower bound of supervised robust learning. In Theorem 5.3, we show that for any $\gamma > 0$ there exists a hypothesis class, such that having access only to $\mathcal{O}(\mathrm{VC}_{\mathcal{U}})$ labeled examples, leads to an error $(\frac{3}{2} - \gamma)\eta + \epsilon$. (All proofs for this section are in Appendix D.)

We start with the upper bounds. First, we analyze the case of using a generic agnostic robust learner, then we deduce the sample complexity of a specific instantiation of such an algorithm.

**Theorem 5.1** *For any hypothesis class $\mathcal{H}$ and adversary $\mathcal{U}$, Algorithm GRASS $(3, \epsilon, \delta)$- PAC learns $\mathcal{H}$ with respect to the robust loss function, in the agnostic robust case, with samples of size*

$$m_l = \mathcal{O}\left(\frac{\mathrm{VC}_{\mathcal{U}}(\mathcal{H})}{\epsilon^2} \log^2 \frac{\mathrm{VC}_{\mathcal{U}}(\mathcal{H})}{\epsilon^2} + \frac{\log\frac{1}{\delta}}{\epsilon^2}\right), \;\; m_u = \Lambda_{\mathrm{AG}}\left(1, \frac{\epsilon}{3}, \frac{\delta}{2}, \mathcal{H}, \mathcal{U}, 2\eta + \frac{\epsilon}{3}\right),$$

*where $\Lambda_{\mathrm{AG}}\left(\alpha, \epsilon, \delta, \mathcal{H}, \mathcal{U}, \eta\right)$ is the sample complexity of adversarially-robust agnostic supervised learning, such that $\eta$ is the error of the optimal hypothesis in $\mathcal{H}$, namely $\eta = \mathrm{R}_{\mathcal{U}}(\mathcal{H}; \mathcal{D})$.*

By using the agnostic supervised robust learner suggested by Montasser et al. [40], we have the following upper bound on the unlabeled sample size, $m_u = \tilde{\mathcal{O}}\left(\frac{\mathrm{VC}(\mathcal{H})\,\mathrm{VC}^*(\mathcal{H})}{\epsilon^2} + \frac{\log\frac{1}{\delta}}{\epsilon^2}\right)$.

We now present two negative results.

**Theorem 5.2** *For any $\epsilon, \delta \in (0, 1)$, the sample complexity of agnostic robust $(1, \epsilon, \delta)$-PAC learning for a class $\mathcal{H}$, with respect to perturbation function $\mathcal{U}$ is (even if $\mathcal{D}_{\mathcal{X}}$ is known),*

$$m_l = \Omega\left(\frac{\mathrm{RS}_{\mathcal{U}}(\mathcal{H})}{\epsilon^2} + \frac{1}{\epsilon^2}\log\frac{1}{\delta}\right) \,, \; m_u = \infty.$$

**Theorem 5.3** *For any $\gamma > 0$, there exists a hypothesis class $\mathcal{H}$ and adversary $\mathcal{U}$, such that the sample complexity for $(\frac{3}{2} - \gamma, \epsilon, \delta)$-PAC learn $\mathcal{H}$ is*

$$m_l = \Omega\left(\frac{\text{VC}_{\mathcal{U}}(\mathcal{H})}{\epsilon^2} + \frac{1}{\epsilon^2}\log\frac{1}{\delta}\right) \ , \ m_u = \infty.$$

**Open question.** What is the optimal error rate in the agnostic setting when using only $\mathcal{O}(\text{VC}_{\mathcal{U}})$ labeled examples?

## 6 Learning with the 0-1 loss assuming robust realizability

In this section, we learn with respect to the 0-1 loss, under robust realizability assumption. A Distribution $\mathcal{D}$ over $\mathcal{X} \times \mathcal{Y}$ is robustly realizable by $\mathcal{H}$ given a perturbation function $\mathcal{U}$, if there is $h \in \mathcal{H}$ such that not only $h$ classifies all points in $\mathcal{D}$ correctly, it also does so with respect to the robust loss function, that is, $\text{R}_{\mathcal{U}}(\mathcal{H}; \mathcal{D}) = 0$. Note that our guarantees, only in this section, are with respect to the non-robust risk. The formal definition is in Appendix E. A simple example of this model is the following. Let $\mathcal{H}$ be linear separators on $\mathcal{X}$ the unit ball in $\mathbb{R}^d$, and $\mathcal{U}$ as $\ell_2$ balls of radius $\gamma$, the robustly realizable distributions are separable with margin $\gamma$, where $\text{VC}_{\mathcal{U}}(\mathcal{H}) = \frac{1}{\gamma^2}$ but $\text{VC}(\mathcal{H}) = d + 1$ can be arbitrarily larger. Moreover, we have the following example. (All proofs are in appendix Appendix E.)

**Proposition 6.1** *For any $m \in \mathbb{N}$, there exist a hypothesis class $\mathcal{H}_m$ and distribution $\mathcal{D}$, such that $\mathcal{D}$ is robustly realizable by $\mathcal{H}_m$, $\text{VC}_{\mathcal{U}}(\mathcal{H}_m) = 1$, and $\text{VC}(\mathcal{H}_m) = 2m$.*

Standard VC theory does not ensure learning in this case. In this section, we explain how we can learn in such a scenario with a small sample complexity (scales linearly in $\text{VC}_{\mathcal{U}}$). Moreover, we show that it cannot be achieved via proper learners.

**Theorem 6.2** *The sample complexity for learning a hypothesis class $\mathcal{H}$ with respect to the 0-1 loss, for any distribution $\mathcal{D}$ that is robustly realizable by $\mathcal{H}$, namely $\text{R}_{\mathcal{U}}(\mathcal{H}; D) = 0$,*

$$\mathcal{O}\left(\frac{\text{VC}_{\mathcal{U}}(\mathcal{H})}{\epsilon}\log^2\frac{\text{VC}_{\mathcal{U}}(\mathcal{H})}{\epsilon} + \frac{\log\frac{1}{\delta}}{\epsilon}\right), \Omega\left(\frac{\text{VC}_{\mathcal{U}}(\mathcal{H})}{\epsilon} + \frac{\log\frac{1}{\delta}}{\epsilon}\right).$$

This Theorem was an intermediate step in the proof of Theorem 4.2, and the sample complexity is the same as Theorem C.1, $\mathcal{O}\left(\Lambda_{\text{RE}}(\epsilon, \delta, \mathcal{H})\right)$. We show that there exists a robust ERM that fails in this setting (Proposition E.2 in Appendix E). Then, we claim that every proper learner fails.

**Theorem 6.3** *There exists $\mathcal{H}$ with $\text{VC}_{\mathcal{U}}(\mathcal{H}) = 1$, such that for any proper learning rule $\mathcal{A}: (\mathcal{X} \times \mathcal{Y})^* \to \mathcal{H}$, there exists a distribution $\mathcal{D}$ over $\mathcal{X} \times \mathcal{Y}$ that is robustly realizable by $\mathcal{H}$, i.e., $\text{R}_{\mathcal{U}}(\mathcal{H}; \mathcal{D}) = 0$, and it holds that $\text{R}(\mathcal{A}(S); D) > \frac{1}{8}$ with probability at least $\frac{1}{7}$ over $S \sim \mathcal{D}^m$.*

## Acknowledgments

We are grateful to Omar Montasser for his helpful input, particularly inspiring steps 3 and 4 of the GRASS learning algorithm. We would like to thank Vinod Raman for his enlightening comments regarding the correctness of our algorithm. Finally, we thank the anonymous reviewers for their thoughtful comments, which helped us improve the presentation of our paper.

This project has received funding from the European Research Council (ERC) under the European Union's Horizon 2020 research and innovation program (grant agreement No. 882396), by the Israel Science Foundation (grants 993/17, 1602/19), Tel Aviv University Center for AI and Data Science (TAD), and the Yandex Initiative for Machine Learning at Tel Aviv University. I.A. is supported by the Vatat Scholarship from the Israeli Council for Higher Education and by the Kreitman School of Advanced Graduate Studies.

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
