# A    Additional preliminaries for Section 2

**Complexity measures.**    The capacity measures, $\mathrm{VC}_{\mathcal{U}}$, $\mathrm{RS}_{\mathcal{U}}$ and VC, play an important role in our results. See Definitions 1.1 and 1.2 for the $\mathrm{VC}_{\mathcal{U}}$ and $\mathrm{RS}_{\mathcal{U}}$ dimensions. It holds that $\mathrm{VC}_{\mathcal{U}}(\mathcal{H}) \leq \mathrm{RS}_{\mathcal{U}}(\mathcal{H}) \leq \mathrm{VC}(\mathcal{H})$, in Proposition 3.2 we demonstrate an arbitrary gap between $\mathrm{VC}_{\mathcal{U}}$ and $\mathrm{RS}_{\mathcal{U}}$, the key parameters controlling the sample complexity of robust learnability.

Denote the projection of a hypothesis class $\mathcal{H}$ on set $S = \{x_1, \ldots, x_k\}$ by $\mathcal{H}|_S = \{(h(x_1), \ldots, h(x_k)) : h \in \mathcal{H}\}$. We say that a set $S \subseteq \mathcal{X}$ is shattered by $\mathcal{H}$ if $\{0, 1\}^S = \mathcal{H}|_S$, the VC-dimension [53] of $\mathcal{H}$ is defined as the maximal size of a shattered set $S$. The dual hypothesis class $\mathcal{H}^* \subseteq \{0, 1\}^{\mathcal{H}}$ is defined as the set of all functions $f_x : \mathcal{H} \to \{0, 1\}$ where $f_x(h) = h(x)$. We denote the VC-dimension of the dual class by $\mathrm{VC}^*(\mathcal{H})$. It is known that $\mathrm{VC}^*(\mathcal{H}) < 2^{\mathrm{VC}(\mathcal{H})+1}$ [5].

**Definition A.1 (Sample compression scheme)**  A pair of functions $(\kappa, \rho)$ is a sample compression scheme of size $\ell$ for class $\mathcal{H}$ if for any $n \in \mathbb{N}$, $h \in \mathcal{H}$ and sample $S = \{(x_i, h(x_i))\}_{i=1}^n$, it holds for the compression function that $\kappa(S) \subseteq S$ and $|\kappa(S)| \leq \ell$, and the reconstruction function $\rho(\kappa(S)) = \hat{h}$ satisfies $\hat{h}(x_i) = h(x_i)$ for any $i \in [n]$.

**Partial concept classes - [2].**    Let a partial concept class $\mathcal{H} \subseteq \{0, 1, \star\}^{\mathcal{X}}$. For $h \in \mathcal{H}$ and input $x$ such that $h(x) = \star$, we say that $h$ is undefined on $x$. The support of a partial hypothesis $h : \mathcal{X} \to \{0, 1, \star\}$ is the preimage of $\{0, 1\}$, formally, $h^{-1}(\{0, 1\}) = \{x \in \mathcal{X} : h(x) \neq \star\}$. The main motivation for introducing partial concept classes is that data-dependent assumptions can be modeled in a natural way that extends the classic theory of total concepts.

The VC-dimension of a partial class $\mathcal{H}$ is defined as the maximum size of a shattered set $S \subseteq \mathcal{X}$, where $S$ is shattered by $\mathcal{H}$ if the projection of $\mathcal{H}$ on $S$ contains all possible binary patterns, $\{0, 1\}^S \subseteq \mathcal{H}|_S$. The VC-dimension also characterizes verbatim the PAC learnability of partial concept classes. However, the uniform convergence argument does not hold, and the ERM principle does not ensure learning. The proof hinges on a combination of sample compression scheme and a variant of the *one-Inclusion-Graph* algorithm [33]. In Section 4 we elaborate on the sample complexity of partial concept classes, and in Appendix F we elaborate on the learning algorithms. The definitions of realizability and agnostic learning in the sense of partial concepts generalize the classic definitions for total concept classes. See [2, Section 2 and Appendix C] for more details.

# B    Proofs for Section 3

**Proof of Proposition 3.2**  We overview the construction by Montasser et al. [40], which exemplifies an arbitrarily large gap between $\mathrm{VC}_{\mathcal{U}}$ and $\mathrm{RS}_{\mathcal{U}}$. In this example $\mathrm{VC}_{\mathcal{U}}(\mathcal{H}) = 0$, $\mathrm{RS}_{\mathcal{U}}(\mathcal{H}) = \infty$, and $\mathrm{VC}(\mathcal{H}) = \infty$.

Define the Euclidean ball of radius $r$ perturbation function $\mathcal{U}(x) = B_r(x)$. Consider infinite sequences $(x_n)_{n \in \mathbb{N}}$ and $(z_n)_{n \in \mathbb{N}}$ of points such that $\forall i \neq j$, $\mathcal{U}(x_i) \cap \mathcal{U}(x_j) = U(x_i) \cap \mathcal{U}(z_j) = \mathcal{U}(x_j) \cap \mathcal{U}(z_i) = \emptyset$, and $\forall i$, $|\mathcal{U}(x_i) \cap \mathcal{U}(z_i)| = 1$.

For a bit string $b \in \{0, 1\}^{\mathbb{N}}$, define a hypothesis $h_b : \{\mathcal{U}(x_i) \cup \mathcal{U}(z_i)\}_{i \in \mathbb{N}} \to \{0, 1\}$ as follows.

$$h_b = \begin{cases} h_b\Big(\mathcal{U}(x_i)\Big) = 1 \ \wedge \ h_b\Big(\mathcal{U}(z_i) \setminus \mathcal{U}(x_i)\Big) = -1, & b_i = 0 \\ h_b\Big(\mathcal{U}(z_i)\Big) = 1 \ \wedge \ h_b\Big(\mathcal{U}(x_i) \setminus \mathcal{U}(z_i)\Big) = -1, & b_i = 1. \end{cases}$$

Define the hypothesis class $\mathcal{H} = \left\{h_b : b \in \{0, 1\}^{\mathbb{N}}\right\}$. It holds that $\mathrm{VC}_{\mathcal{U}}(\mathcal{H}) = 0$ and $\mathrm{RS}_{\mathcal{U}} = \infty$. ∎

# C    Proofs for Section 4

Before proceeding to the proof, we present the following result on learning partial concept classes. Recall the definition of VC is in the context of partial concepts (see Appendix A).

**Theorem C.1 ([2], Theorem 34)** *Any partial concept class $\mathcal{H}$ with $\text{VC}(\mathcal{H}) < \infty$ is PAC learnable in the realizable setting with sample complexity,*

- $\Lambda_{\text{RE}}(\epsilon, \delta, \mathcal{H}) = \mathcal{O}\left(\min\left\{\frac{\text{VC}(\mathcal{H})}{\epsilon}\log\frac{1}{\delta}, \frac{\text{VC}(\mathcal{H})}{\epsilon}\log^2\left(\frac{\text{VC}(\mathcal{H})}{\epsilon}\right) + \frac{1}{\epsilon}\log\frac{1}{\delta}\right\}\right)$

- $\Lambda_{\text{RE}}(\epsilon, \delta, \mathcal{H}) = \Omega\left(\frac{\text{VC}(\mathcal{H})}{\epsilon} + \frac{1}{\epsilon}\log\frac{1}{\delta}\right)$.

**Proof of Theorem 4.2** At first, we convert the hypothesis class $\mathcal{H}$ to $\mathcal{H}_{\mathcal{U}}^{\star}$ as described in Definition 4.1. Then, we employ the learning algorithm $\mathcal{A}$ for partial concepts on the partial concept class $\mathcal{H}_{\mathcal{U}}^{\star}$ and $S^l$, denote the resulting hypothesis by $h_1$. Note that we reduced the complexity of the class, since $\text{VC}(\mathcal{H}_{\mathcal{U}}^{\star}) = \text{VC}_{\mathcal{U}}(\mathcal{H})$. Theorem C.1 implies that whenever $m_l = |S^l| \geq \tilde{\mathcal{O}}\left(\frac{\text{VC}_{\mathcal{U}}(\mathcal{H})}{\epsilon} + \frac{1}{\epsilon}\log\frac{1}{\delta}\right)$, the hypothesis $h_1$ has a non-robust error at most $\frac{\epsilon}{3}$ with probability $1 - \frac{\delta}{2}$, with respect to the 0-1 loss. Note that there exists $h \in \mathcal{H}$ that classifies correctly any point in $\mathcal{D}$ with respect to the robust loss function. So when we convert $\mathcal{H}$ to $\mathcal{H}_{\mathcal{U}}^{\star}$, the "partial version" of $h$ still classifies correctly any point in $S^l$, and does not return any $\star$, which always counts as a mistake. Algorithm $\mathcal{A}$ guarantees to return a hypothesis that is $\epsilon$-optimal with respect to the 0-1 loss, with high probability. Observe that after these two steps, we obtain the following intermediate result. Whenever a distribution $\mathcal{D}$ is robustly realizable by a hypothesis class $\mathcal{H}$, i.e., $\text{R}_{\mathcal{U}}(\mathcal{H}; \mathcal{D}) = 0$, we have an algorithm that learns this class with respect to the 0-1 loss, with sample complexity of

$$\Upsilon(\epsilon, \delta, \mathcal{H}, \mathcal{U}) = \mathcal{O}(\Lambda_{\text{RE}}(\epsilon, \delta, \mathcal{H})) = \mathcal{O}\left(\frac{\text{VC}_{\mathcal{U}}(\mathcal{H})}{\epsilon}\log^2\frac{\text{VC}_{\mathcal{U}}(\mathcal{H})}{\epsilon} + \frac{1}{\epsilon}\log\frac{1}{\delta}\right). \tag{1}$$

The sample complexity of this model is defined formally in Definition E.1. See Section 6 for more results for this model.

In the third step, we label an independent unlabeled sample $S_{\mathcal{X}}^u \sim \mathcal{D}_{\mathcal{X}}^{m_u}$ with $h_1$, denote this labeled sample by $S^u$. Define a distribution $\tilde{\mathcal{D}}$ over $\mathcal{X} \times \mathcal{Y}$ by

$$\tilde{\mathcal{D}}(x, h_1(x)) = \mathcal{D}_{\mathcal{X}}(x),$$

and so $S^u$ is an i.i.d. sample from $\tilde{\mathcal{D}}$. We argue that the robust error of $\mathcal{H}$ with respect to $\tilde{\mathcal{D}}$ is at most $\frac{\epsilon}{3}$, i.e., $\text{R}_{\mathcal{U}}(\mathcal{H}; \tilde{\mathcal{D}}) \leq \frac{\epsilon}{3}$. Indeed, we show that $h_{\text{opt}} \in \arg\min_{h \in \mathcal{H}} \text{R}_{\mathcal{U}}(h; \mathcal{D})$ has a robust error of at most $\frac{\epsilon}{3}$ on $\tilde{\mathcal{D}}$. Note that,

$$\text{R}_{\mathcal{U}}(\mathcal{H}; \tilde{\mathcal{D}}) \leq \mathbb{E}_{(x,y)\sim\mathcal{D}}\left[\ell_{\mathcal{U}}(h_{\text{opt}}; x, h_1(x))\right] = \mathbb{E}_{(x,y)\sim\tilde{\mathcal{D}}}\left[\ell_{\mathcal{U}}(h_{\text{opt}}; x, y)\right]. \tag{2}$$

Observe that the following holds for any $(x, y)$,

$$\ell_{\mathcal{U}}(h_{\text{opt}}; x, h_1(x)) \leq \ell_{\mathcal{U}}(h_{\text{opt}}; x, y) + \ell_{\text{0-1}}(h_1; x, y). \tag{3}$$

Indeed, the right-hand side is 0, whenever $h_1$ classifies $(x, y)$ correctly, and $h_{\text{opt}}$ robustly classifies $(x, y)$ correctly, which implies that the left-hand side is 0 as well.

By taking the expectation on Eq. (3) we have,

$$\mathbb{E}_{(x,y)\sim\mathcal{D}}[\ell_{\mathcal{U}}(h_{\text{opt}}; x, h_1(x))] \leq \mathbb{E}_{(x,y)\sim\mathcal{D}}[\ell_{\mathcal{U}}(h_{\text{opt}}; x, y)] + \mathbb{E}_{(x,y)\sim\mathcal{D}}[\ell_{\text{0-1}}(h_1; x, y)]. \tag{4}$$

We have

$$\begin{aligned}
\text{R}_{\mathcal{U}}(\mathcal{H}; \tilde{\mathcal{D}}) &\leq \mathbb{E}_{(x,y)\sim\tilde{\mathcal{D}}}\left[\ell_{\mathcal{U}}(h_{\text{opt}}; x, y)\right] \\
&\overset{(i)}{=} \mathbb{E}_{(x,y)\sim\mathcal{D}}[\ell_{\mathcal{U}}(h_{\text{opt}}; x, h_1(x))] \\
&\overset{(ii)}{\leq} \mathbb{E}_{(x,y)\sim\mathcal{D}}[\ell_{\mathcal{U}}(h_{\text{opt}}; x, y)] + \mathbb{E}_{(x,y)\sim\mathcal{D}}[\ell_{\text{0-1}}(h_1; x, y)] \\
&\leq \frac{\epsilon}{3}
\end{aligned}$$

where (i) follows from Eq. (2) and (ii) follows from Eq. (4).

Finally, we employ an agnostic adversarially robust supervised PAC learner $\mathcal{B}$ for the class $\mathcal{H}$ on $S^u \sim \tilde{\mathcal{D}}^{m_u}$, that should be of a size of the sample complexity of agnostically robust learn $\mathcal{H}$ with respect to $\mathcal{U}$, when the optimal robust error of hypothesis from $\mathcal{H}$ on $\tilde{\mathcal{D}}$ is at most $\frac{\epsilon}{3}$. We are guaranteed that the resulting hypothesis $h_2$ has a robust error of at most $\frac{\epsilon}{3} + \frac{\epsilon}{3} = \frac{2\epsilon}{3}$ on $\tilde{\mathcal{D}}$, with probability $1 - \frac{\delta}{2}$. We observe that the total variation distance between $\mathcal{D}$ and $\tilde{\mathcal{D}}$ is at most $\frac{\epsilon}{3}$, and as a result, $h_2$ has a robust error of at most $\frac{2\epsilon}{3} + \frac{\epsilon}{3} = \epsilon$ on $\mathcal{D}$, with probability $1 - \delta$.

We conclude that a size of $|S_{\mathcal{X}}^u| = m_u = \Lambda_{\text{AG}}\left(1, \frac{\epsilon}{3}, \frac{\delta}{2}, \mathcal{H}, \mathcal{U}, \eta = \frac{\epsilon}{3}\right)$ unlabeled samples suffices, in addition to $m_l = \tilde{\mathcal{O}}\left(\frac{\text{VC}_{\mathcal{U}}(\mathcal{H})}{\epsilon} + \frac{1}{\epsilon}\log\frac{1}{\delta}\right)$ labeled samples which are required in the first 2 steps. ∎

We now prove Theorem 4.4. The following data-dependent compression-based generalization bound is a variation of the classic bound by Graepel et al. [28]. It follows the same arguments while using the empirical Bernstein bound instead of Hoeffding's inequality. A variation of this bound, with respect to the 0-1 loss, appears in [2, Lemma 42], and [38, Section 5]. The exact same arguments follow for the robust loss as well.

This bound includes the empirical error factor, and as soon as we call the compression-based learner on a sample that is "nearly" realizable (Step 4 in the algorithm), we can improve the sample complexity of the agnostic robust supervised learner, such that the dependence on $\epsilon^2$ is reduced to $\epsilon$, for the unlabeled sample size.

**Lemma C.2 (Agnostic sample compression generalization bound)** *For any sample compression scheme $(\kappa, \rho)$, for any $m \in \mathbb{N}$ and $\delta \in (0, 1)$, for any distribution $\mathcal{D}$ over $\mathcal{X} \times \{0, 1\}$, for $S \sim \mathcal{D}^m$, with probability $1 - \delta$,*

$$\left|\text{R}_{\mathcal{U}}\left(\rho(\kappa(S)); \mathcal{D}\right) - \widehat{\text{R}}_{\mathcal{U}}\left(\rho(\kappa(S)); S\right)\right| \leq \mathcal{O}\left(\sqrt{\widehat{\text{R}}_{\mathcal{U}}\left(\rho(\kappa(S)); S\right) \frac{\left(|\kappa(S)|\log(m) + \log\frac{1}{\delta}\right)}{m}} + \frac{|\kappa(S)|\log(m) + \log\frac{1}{\delta}}{m}\right).$$

**Proof of Theorem 4.4** Montasser et al. [40, Theorem 6] introduced an agnostic robust supervised learner that requires the following labeled sample size,

$$\Lambda_{\text{AG}}\left(1, \epsilon, \delta, \mathcal{H}, \mathcal{U}, \eta\right) = \tilde{\mathcal{O}}\left(\frac{\text{VC}(\mathcal{H})\,\text{VC}^*(\mathcal{H})}{\epsilon^2} + \frac{\log\frac{1}{\delta}}{\epsilon^2}\right).$$

Their argument for generalization is based on classic compression generalization bound by Graepel et al. [28], adapted to the robust loss. See Montasser et al. [40, Lemma 11].

We show that in our use case, we can deduce a stronger bound. We employ the agnostic learner on a distribution that is "close" to realizable, the error of the optimal $h \in \mathcal{H}$ is at most $\eta = \frac{\epsilon}{3}$, and so we need $\Lambda_{\text{AG}}\left(1, \frac{\epsilon}{3}, \frac{\delta}{2}, \mathcal{H}, \mathcal{U}, \eta = \frac{\epsilon}{3}\right)$ unlabeled examples. As a result, we obtain an improved bound by using a data-dependant generalization bound described in Lemma C.2.

This improves the unlabeled sample size (denoted by $m_u$) and reduces its dependence on $\epsilon^2$ to $\epsilon$. Overall we obtain a sample complexity of

$$m_u = \tilde{\mathcal{O}}\left(\frac{\text{VC}(\mathcal{H})\,\text{VC}^*(\mathcal{H})}{\epsilon} + \frac{\log\frac{1}{\delta}}{\epsilon}\right), \quad m_l = \mathcal{O}\left(\frac{\text{VC}_{\mathcal{U}}(\mathcal{H})}{\epsilon}\log^2\frac{\text{VC}_{\mathcal{U}}(\mathcal{H})}{\epsilon} + \frac{\log\frac{1}{\delta}}{\epsilon}\right).$$

∎

**Proof of Theorem 4.6** This proof is identical to [40, Lemma 3], We overview the idea of the proof. If the proof is true for a labeled sample, it remains true when some of the labels are missing.

Define the following hypothesis class $\mathcal{H}_m \subseteq [0, 1]^{\mathcal{X}}$. Define the instance space $\mathcal{X} = \{x_1, \dots, x_m\} \subseteq \mathbb{R}$ and a perturbation function $\mathcal{U} : \mathcal{X} \to 2^{\mathcal{X}}$, such that the perturbation sets of the instances do not intersect, that is, $\forall i, j \in [m] : \mathcal{U}(x_i) \cap \mathcal{U}(x_j)$. We can simply take the perturbations sets to be $\ell_2$ unit balls, $\mathcal{U}(x) = \{z \in \mathbb{R} : \|z - x\|_2 \leq 1\}$ such that $\forall i, j \in [m] : \|x_i - x_j\|_2 > 2$. Now, each

$h_b \in \mathcal{H}_m$ is represented by a bit string $b = \{0,1\}^m$, such that if $b_i = 1$, then there exist an adversarial example in $\mathcal{U}(x_i)$ that is unique for each $h_b$, and otherwise, the function is consistent on $\mathcal{U}(x_i)$.

Formally, for each $i \in [m]$ define a bijection $\psi_i : x_i \times \mathcal{H}_m \to \mathcal{U}(x_i) \setminus \{x_i\}$. Define $\mathcal{H}_m = \{h_b : b \in \{0,1\}^m\}$, such that for any $x_i \in \mathcal{X}$, $h_b$ is defined by

$$h_b(x_i) = \begin{cases} h_b\Big(\mathcal{U}(x_i) \setminus \psi_i(x_i, h_b)\Big) = 0 \ \wedge \ h_b\Big(\psi_i(x_i, h_b)\Big) = 1, & b_i = 1, \\ h_b\Big(\mathcal{U}(x_i)\Big) = 0, & b_i = 0. \end{cases}$$

Note that since $\psi_i$ is a bijection, different functions with $b_i = 1$ have a different perturbation for $x_i$ that causes a misclassification.

For a function class $\mathcal{H}$, define the robust loss class $\mathcal{L}_{\mathcal{H}}^{\mathcal{U}} = \Big\{(x,y) \mapsto \sup_{z \in \mathcal{U}(x)} \mathbb{I}\{h(z) \neq y\} : h \in \mathcal{H}\Big\}$. It holds that $\mathrm{VC}(\mathcal{H}_m) \leq 1$ and $\mathrm{VC}(\mathcal{L}_{\mathcal{H}_m}^{\mathcal{U}}) = m$ (see [40, Lemma 2]).

We define a function class $\tilde{\mathcal{H}}_{3m} = \Big\{h_b \in \mathcal{H}_{3m} : \sum_{i=1}^{3m} b_i = m\Big\}$. In words, we are keeping only functions in $\mathcal{H}_{3m}$ that are robustly correct on exactly $2m$ points. Note that the function $h_{\vec{0}}$ (bit string of all zeros) which is robustly correct on all $3m$ points, is not the class.

The idea is that we can construct a family of $\binom{3m}{2m}$ distributions, such that each distribution is supported on $2m$ points from $\mathcal{X} = \{x_1, \ldots, x_{3m}\}$. Now, if we have a proper learning rule, observing only $m$ points, the algorithm has no information which is the remaining $m$ points in the support (out of $2m$ possible points in $\mathcal{X}$). For each such a distribution there exists $h \in \tilde{\mathcal{H}}_{3m}$, with zero robust error. We can follow a standard proof of the no-free-lunch theorem [e.g., 50, Section 5], showing via the probabilistic method, that there exists a distribution on which the algorithm has a constant error, although there is an optimal function in $\tilde{\mathcal{H}}_{3m}$. See [40, Lemma 3] for the full proof. ∎

# D    Proofs for Section 5

Before proceeding to the proof, we present the following result on agnostically learning partial concept classes. Recall the definition of VC is in the context of partial concepts (see Appendix A).

**Theorem D.1 ([2], Theorem 41)** *Any partial concept class $\mathcal{H}$ with $\mathrm{VC}(\mathcal{H}) < \infty$ is agnostically* PAC *learnable with sample complexity,*

- $\Lambda_{\mathrm{AG}}(\epsilon, \delta, \mathcal{H}) = \mathcal{O}\left(\frac{\mathrm{VC}(\mathcal{H})}{\epsilon^2} \log^2\left(\frac{\mathrm{VC}(\mathcal{H})}{\epsilon^2}\right) + \frac{1}{\epsilon^2} \log \frac{1}{\delta}\right)$.

- $\Lambda_{\mathrm{AG}}(\epsilon, \delta, \mathcal{H}) = \Omega\left(\frac{\mathrm{VC}(\mathcal{H})}{\epsilon^2} + \frac{1}{\epsilon^2} \log \frac{1}{\delta}\right)$.

**Proof of Theorem 5.1** We follow the same steps as in the proof of the realizable case, with the following difference. In the first two steps of the algorithm, we learn with respect to the 0-1 loss, with an error of $\eta$ (the optimal robust error of a hypothesis in $\mathcal{H}$) and not 0, which leads eventually to an approximation of $3\eta$ for learning with the robust loss.

At first, we convert the class $\mathcal{H}$ into $\mathcal{H}_{\mathcal{U}}^{\star}$, on which we employ the learning algorithm $\mathcal{A}$ for partial concepts with the sample $S^l$. Theorem D.1 implies that whenever $m_l = |S^l| \geq \tilde{\mathcal{O}}\left(\frac{\mathrm{VC}_{\mathcal{U}}(\mathcal{H})}{\epsilon^2} + \frac{1}{\epsilon^2} \log \frac{1}{\delta}\right)$, the resulting hypothesis $h_1$ returned by algorithm $\mathcal{A}$ has a non-robust error at most $\eta + \frac{\epsilon}{3}$ with probability $1 - \frac{\delta}{2}$, with respect to the 0-1 loss, where $\eta = \mathrm{R}_{\mathcal{U}}(\mathcal{H}; \mathcal{D})$. Note that there exists $h \in \mathcal{H}$ with robust error of $\eta$ on $\mathcal{D}$. The "partial version" of $h$ has an error of $\eta$ on $\mathcal{D}$ with respect to the 0-1 loss. As a result, algorithm $\mathcal{A}$ guarantees to return a hypothesis that is $\epsilon$-optimal with respect to the 0-1 loss, with high probability.

We label an independent unlabeled sample $S^u_\mathcal{X} \sim \mathcal{D}^{m_u}_\mathcal{X}$ with $h_1$, denote this labeled sample by $S^u$. Similarly to the realizable case, define a distribution $\tilde{\mathcal{D}}$ over $\mathcal{X} \times \mathcal{Y}$ by

$$\tilde{\mathcal{D}}(x, h_1(x)) = \mathcal{D}_\mathcal{X}(x),$$

and so $S^u$ is an i.i.d. sample from $\tilde{\mathcal{D}}$. We argue that the robust error of $\mathcal{H}$ with respect to $\tilde{\mathcal{D}}$ is at most $2\eta + \frac{\epsilon}{3}$, i.e., $\mathrm{R}_\mathcal{U}(\mathcal{H}; \tilde{\mathcal{D}}) = 2\eta + \frac{\epsilon}{3}$, by showing that $h_{\mathrm{opt}} = \mathrm{argmin}_{h \in \mathcal{H}} \mathrm{R}_\mathcal{U}(h; \mathcal{D})$ has a robust error of at most $2\eta + \frac{\epsilon}{3}$ on $\tilde{\mathcal{D}}$.

Eqs. (2) to (4) still hold as in the realizable case proof. Combining it together, we have

$$
\begin{aligned}
\mathrm{R}_\mathcal{U}(\mathcal{H}; \tilde{\mathcal{D}}) &\leq \mathbb{E}_{(x,y) \sim \tilde{\mathcal{D}}} \left[ \ell_\mathcal{U}(h_{\mathrm{opt}}; x, y) \right] \\
&\overset{(i)}{=} \mathbb{E}_{(x,y) \sim \mathcal{D}} [\ell_\mathcal{U}(h_{\mathrm{opt}}; x, h_1(x))] \\
&\overset{(ii)}{\leq} \mathbb{E}_{(x,y) \sim \mathcal{D}}[\ell_\mathcal{U}(h_{\mathrm{opt}}; x, y)] + \mathbb{E}_{(x,y) \sim \mathcal{D}}[\ell_{\text{0-1}}(h_1; x, y)] \\
&\leq \eta + \eta + \frac{\epsilon}{3} \\
&= 2\eta + \frac{\epsilon}{3},
\end{aligned}
$$

where (i) follows from Eq. (2) and (ii) follows from Eq. (4).

Finally, we employ an agnostic adversarially robust supervised PAC learner $\mathcal{B}$ for the class $\mathcal{H}$ on $S^u \sim \tilde{\mathcal{D}}^{m_u}$, that should be of a size of the sample complexity of agnostically robust learn $\mathcal{H}$ with respect to $\mathcal{U}$, when the optimal robust error of hypothesis from $\mathcal{H}$ on $\tilde{\mathcal{D}}$ is at most $2\eta + \frac{\epsilon}{3}$. We are guaranteed that the resulting hypothesis $h_2$ has a robust error of at most $2\eta + \frac{\epsilon}{3} + \frac{\epsilon}{3} = 2\eta + \frac{2\epsilon}{3}$ on $\tilde{\mathcal{D}}$, with probability $1 - \frac{\delta}{2}$. We observe that the total variation distance between $\mathcal{D}$ and $\tilde{\mathcal{D}}$ is at most $\eta + \frac{\epsilon}{3}$, and as a result, $h_2$ has a robust error of at most $2\eta + \frac{2\epsilon}{3} + \eta + \frac{\epsilon}{3} = 3\eta + \epsilon$ on $\mathcal{D}$, with probability $1 - \delta$.

We conclude that a size of $|S^u_\mathcal{X}| = m_u = \Lambda_{\mathrm{AG}}\left(1, \frac{\epsilon}{3}, \frac{\delta}{2}, \mathcal{H}, \mathcal{U}, 2\eta + \frac{\epsilon}{3}\right)$ unlabeled sample suffices, in addition to the $m_l = \mathcal{O}\left(\frac{\mathrm{VC}_\mathcal{U}(\mathcal{H})}{\epsilon^2} \log^2 \frac{\mathrm{VC}_\mathcal{U}(\mathcal{H})}{\epsilon^2} + \frac{\log \frac{1}{\delta}}{\epsilon^2}\right)$ labeled samples which are required in the first 2 steps. We remark that the best known value of $\Lambda_{\mathrm{AG}}(1, \epsilon, \delta, \mathcal{H}, \mathcal{U}, \eta)$ is $\tilde{\mathcal{O}}\left(\frac{\mathrm{VC}(\mathcal{H})\, \mathrm{VC}^*(\mathcal{H})}{\epsilon^2} + \frac{\log \frac{1}{\delta}}{\epsilon^2}\right)$. ∎

**Proof of Theorem 5.2** We give a proof sketch, this is similar to [40, Theorem 10], knowing the marginal distribution $\mathcal{D}_\mathcal{X}$ does not give more power to the learner. The argument is based on the standard lower bound for VC classes (for example [39, Section 3]). Let $S = \{x_1, \ldots, x_k\}$ be a maximal set that is $\mathcal{U}$-robustly shattered by $\mathcal{H}$.

Let $z^+_1, z^-_1, \ldots, z^+_k, z^-_k$ be as in Definition 1.2, and note that for $i \neq j$, $z^+_i \neq z^+_j$ and $z^-_i \neq z^-_j$. Define a distribution $\mathcal{D}_{\boldsymbol{\sigma}}$ for any possible labeling $\boldsymbol{\sigma} = (\sigma_1, \ldots, \sigma_k) \in \{0,1\}^k$ of $S$.

$$
\forall j \in [k]: \begin{cases} \mathcal{D}_{\boldsymbol{\sigma}}(z^+_j, 1) = \frac{1-\alpha}{2k} \ \wedge \ \mathcal{D}_{\boldsymbol{\sigma}}(z^-_j, 0) = \frac{1+\alpha}{2k} & \sigma_j = 0, \\ \mathcal{D}_{\boldsymbol{\sigma}}(z^+_j, 1) = \frac{1+\alpha}{2k} \ \wedge \ \mathcal{D}_{\boldsymbol{\sigma}}(z^-_j, 0) = \frac{1-\alpha}{2k} & \sigma_j = 1. \end{cases}
$$

We can now choose $\alpha$ as a function of $\epsilon, \delta$ in order to get a lower bound on the sample complexity $|S| \gtrsim \frac{\mathrm{RS}_\mathcal{U}}{\epsilon^2}$. ∎

**Proof of Theorem 5.3** We take the construction in Proposition 3.2, where there is an arbitrary gap between $\mathrm{VC}_\mathcal{U}$ and $\mathrm{RS}_\mathcal{U}$.

Recall that on every pair $(x, z)$ in Proposition 3.2 the optimal error is $\eta = 1/2$. On such unlabeled pairs, the learner can only randomly choose a prediction, and the error is $3/4$. We have $\mathrm{VC}_\mathcal{U} = 0$,

and the labeled sample size is $\frac{1}{\epsilon^2} \log \frac{1}{\delta}$. As $(\mathrm{RS}_{\mathcal{U}} - \frac{1}{\epsilon^2} \log \frac{1}{\delta})$ grows, the gap between the learner and the optimal classifier is approaching $3/2$, which means that for any $\gamma > 0$ we can pick $\mathrm{RS}_{\mathcal{U}}$ such that error of $(\frac{2}{3} - \gamma)\eta$ is not possible.

In order to prove the case of any $0 < \eta \leq 1/2$, we can just add points such that their perturbation set does not intersect with any other perturbation set, and follow the same argument. ∎

# E   Auxiliary definitions and proofs for Section 6

Definition of the model.

**Definition E.1 ((non-robust)** PAC **learnability for robustly realizable distributions)** For any $\epsilon, \delta \in (0, 1)$, the sample complexity of $(\epsilon, \delta)$-PAC learning for a class $\mathcal{H}$, denoted by $\Upsilon(\epsilon, \delta, \mathcal{H}, \mathcal{U})$, is the smallest integer $m$ for which there exists a learning algorithm $\mathcal{A} : (\mathcal{X} \times \mathcal{Y})^* \to \mathcal{Y}^{\mathcal{X}}$, such that for every distribution $\mathcal{D}$ over $\mathcal{X} \times \mathcal{Y}$ robustly realizable by $\mathcal{H}$ with respect to a perturbation function $\mathcal{U} : \mathcal{X} \to 2^{\mathcal{X}}$, namely $\mathrm{R}_{\mathcal{U}}(\mathcal{H}; D) = 0$, for a random sample $S \sim \mathcal{D}^m$, it holds that

$$\mathbb{P}\left(\mathrm{R}\left(\mathcal{A}(S); D\right) \leq \epsilon\right) > 1 - \delta.$$

If no such $m$ exists, define $\Upsilon(\epsilon, \delta, \mathcal{H}, \mathcal{U}) = \infty$, and $\mathcal{H}$ is not $(\epsilon, \delta)$-PAC for distributions that are robustly realizable by $\mathcal{H}$ with respect to $\mathcal{U}$.

**Proof of Proposition 6.1** Define the uniform distribution $\mathcal{D}$ over the support $\{(x_1, 1), \dots, (x_{2m}, 1)\}$, such that $\bigcap_{i=1}^{2m} \mathcal{U}(x_i) \neq \emptyset$. Define $\mathcal{H} : \mathcal{X} \to 2^{\mathcal{X}}$ to be all binary functions over $\mathcal{X}$. Note that the $\mathcal{D}$ is robustly realizable by $\mathcal{H}$, the constant function that returns always 1 has no error. Moreover we have $\mathrm{VC}_{\mathcal{U}} = 1$, and $\mathrm{VC} = 2m$, for any $m \in \mathbb{N}$. ∎

**Proof of Theorem 6.2** We follow only the first two steps of the generic Algorithm 1. Namely, take a labeled sample $S$ and a hypothesis class $\mathcal{H}$ and create the partial hypothesis class $\mathcal{H}_{\mathcal{U}}^{\star}$. Assuming that the distribution is robustly realizable by $\mathcal{H}$, we end up in a realizable setting of learning a partial concept class $\mathcal{H}_{\mathcal{U}}^{\star}$.

In the second step of the algorithm, we call a learning algorithm for partial concept classes (Appendix F) in order to do so. The sample complexity is the same as Theorem C.1, $\Upsilon(\epsilon, \delta, \mathcal{H}, \mathcal{U}) = \mathcal{O}\left(\Lambda_{\mathrm{RE}}(\epsilon, \delta, \mathcal{H})\right)$. The Theorem follows from Eq. (1) in Theorem 4.2. ∎

**Proposition E.2** *Consider the distribution $\mathcal{D}$ and the hypothesis class $\mathcal{H}$ in Proposition 6.1. There exists a robust ERM algorithm returning a hypothesis $h_{ERM} \in \mathcal{H}$, such that $\mathrm{R}(h_{ERM}; \mathcal{D}) \geq \frac{1}{4}$ with probability 1 over $S \sim \mathcal{D}^m$.*

**Proof** Consider the following robust ERM. For any sample of size $m$, return 1 on the sample points and randomly choose a label for out-of-sample points. The error rate of such a robust ERM is at least $1/4$ with probability 1. ∎

**Proof of Theorem 6.3** This follows from a similar no-free-lunch argument for VC classes [e.g., 50, Section 5]. We briefly explain the proof idea.

Take the distribution $\mathcal{D}$, and the class $\mathcal{H}$ from Proposition E.2 with $\mathrm{VC}_{\mathcal{U}}(\mathcal{H}) = 1$ and $\mathrm{VC}(\mathcal{H}) = 3m$. Keep functions that are robustly self-consistent only on $2m$ points. Construct all distributions on $2m$ points from the support of $\mathcal{D}$. We have $\binom{3m}{2m}$ such distributions, and each one of them is robustly realizable by different $h \in \mathcal{H}$. The idea is that a proper leaner observing only $m$ points should guess

the remaining $m$ points in the support of the distribution. The rest of the proof follows from the no-free-lunch proof. It can be shown formally via the probabilistic method, that for every proper rule, there exists a distribution on which the error is constant with a fixed probability. ∎

# F   Learning algorithms for partial concept classes

Here we overview the algorithmic techniques from Alon et al. [2, Theorem 34 and 41], for learning partial concepts in realizable and agnostic settings. We use these algorithms in step 2 of our Algorithm 1.

**One-inclusion graph algorithm for partial concept classes.**   We briefly discuss the algorithm, for the full picture, see [54, 33]. The one-inclusion algorithm for a class $\mathcal{F} \subseteq \{0, 1, \star\}^{\mathcal{X}}$ gets an input of unlabeled examples $S = (x_1, \ldots, x_m)$ and labels $(y_1, \ldots, y_{i-1}, y_{i+1}, \ldots, y_m)$ that are consistent with some $f \in \mathcal{F}$, that is, $f(x_k) = y_k$ for all $k \neq i$. It guarantees an $(\epsilon, \delta)$- PAC learner in the realizable setting, with sample complexity of $\Lambda_{\mathrm{RE}}(\epsilon, \delta, \mathcal{H}) = \mathcal{O}\left(\frac{\mathrm{VC}(\mathcal{H})}{\epsilon} \log \frac{1}{\delta}\right)$ as mentioned in Theorem C.1.

Here is a description of the algorithm. First, construct the one-inclusion graph. For any $j \in [m]$ and $f \in \mathcal{F}|_S$ define $E_{j,f} = \{f' \in \mathcal{F}|_S : f'(x_k) = f(x_k), \forall k \neq j\}$, that is, all functions in $\mathcal{F}|_S$ that are consistent with $f$ on $S$, except the point $x_j$. Define the set of edges $E = \{E_{j,f} : j \in [m], f \in \mathcal{F}|_S\}$, and the set vertices $V = \mathcal{F}|_S$ of the one-inclusion graph $G = (V, E)$. An orientation function $\psi : E \to V$ for an undirected graph $G$ is an assignment of a direction to each edge, turning $G$ into a directed graph. Find an orientation $\psi$ that minimizes the out-degree of $G$. For prediction of $x_i$, pick $f \in V$ such that $f(x_k) = y_k$ for all $k \neq i$, and output $\psi(E_{i,f})(x_i)$.

Note that this algorithm is transductive, in the sense that in order to predict the label of a test point, it uses the entire training sample to compute its prediction.

**Boosting and compression schemes.**   Recall the well-known boosting algorithm, $\alpha$-Boost [48, pages 162-163], which is a simplified version of AdaBoost, where the returned function is a simple majority over weak learners, instead of a weighted majority. For a hypothesis class $\mathcal{H}$ and a sample of size $m$, the algorithm yields a compression scheme of size $\mathcal{O}(\mathrm{VC}(\mathcal{H}) \log(m))$. Recall the following generalization bound based on a sample compression scheme.

**Lemma F.1 ([28])** *Let a sample compression scheme* $(\kappa, \rho)$, *and a loss function* $\ell : \mathbb{R} \times \mathbb{R} \to [0, 1]$. *In the agnostic case, for any* $\kappa(S) \lesssim m$, *any* $\delta \in (0, 1)$, *and any distribution* $\mathcal{D}$ *over* $\mathcal{X} \times \{0, 1\}$, *for* $S \sim \mathcal{D}^m$, *with probability* $1 - \delta$,

$$\left| \mathrm{R}\left(\rho(\kappa(S)); \mathcal{D}\right) - \widehat{\mathrm{R}}\left(\rho(\kappa(S)); S\right) \right| \leq \mathcal{O}\left( \sqrt{\frac{\left(|\kappa(S)| \log(m) + \log \frac{1}{\delta}\right)}{m}} \right).$$

The learning algorithm for the realizable setting is $\alpha$-Boost, where the weak learners are taken from the one-inclusion graph algorithm. As mentioned in Theorem C.1, this obtains an upper bound of $\Lambda_{\mathrm{RE}}(\epsilon, \delta, \mathcal{H}) = \mathcal{O}\left(\frac{\mathrm{VC}(\mathcal{H})}{\epsilon} \log^2\left(\frac{\mathrm{VC}(\mathcal{H})}{\epsilon}\right) + \frac{1}{\epsilon} \log \frac{1}{\delta}\right)$.

For the agnostic setting, follow a reduction to the realizable case suggested by David et al. [24], which is based on a construction of a compression scheme. Roughly speaking, the reduction works as follows. Denote $\Lambda_{\mathrm{RE}} = \Lambda_{\mathrm{RE}}(1/3, 1/3, \mathcal{H})$, the sample complexity of $(1/3, 1/3)$-PAC learn $\mathcal{H}$, in the realizable case. Now, $\Lambda_{\mathrm{RE}}$ samples suffice for weak learning for any distribution $\mathcal{D}$ on a given sample $S$.

Find the maximal subset $S' \subseteq S$ such that $\inf_{h \in \mathcal{H}} \widehat{\mathrm{R}}(h; S') = 0$. Now, $\Lambda_{\mathrm{RE}}$ samples suffice for weak robust learning for any distribution $\mathcal{D}$ on $S'$. Execute the $\alpha$-boost algorithm on $S'$, with parameters $\alpha = \frac{1}{3}$ and number of boosting rounds $T = \mathcal{O}(\log(|S'|))$, where each weak learner is trained on $\Lambda_{\mathrm{RE}}$ samples. The returned hypothesis $\bar{h} = \mathrm{Majority}\left(\hat{h}_1, \ldots, \hat{h}_T\right)$ satisfies that $\widehat{\mathrm{R}}(\bar{h}; S') = 0$,

and each hypothesis $\hat{h}_t \in \left\{ \hat{h}_1, \ldots, \hat{h}_T \right\}$ is representable as set of size $\mathcal{O}(\Lambda_{\mathrm{RE}})$. This defines a compression scheme of size $\Lambda_{\mathrm{RE}}T$, and $\bar{h}$ can be reconstructed from a compression set of points from $S$ of size $\Lambda_{\mathrm{RE}}T$.

Recall that $S' \subseteq S$ is a maximal subset such that $\inf_{h \in \mathcal{H}} \widehat{\mathrm{R}}\left(h; S'\right) = 0$ which implies that $\widehat{\mathrm{R}}\left(\bar{h}; S\right) \leq \inf_{h \in \mathcal{H}} \widehat{\mathrm{R}}\left(h; S\right)$. Plugging it into a data-dependent compression generalization bound (Lemma C.2), we obtain a sample complexity of $\Lambda_{\mathrm{AG}}\left(\epsilon, \delta, \mathcal{H}\right) = \mathcal{O}\left( \frac{\mathrm{VC}(\mathcal{H})}{\epsilon^2} \log^2\left( \frac{\mathrm{VC}(\mathcal{H})}{\epsilon^2} \right) + \frac{1}{\epsilon^2} \log \frac{1}{\delta} \right)$, as mentioned in Theorem D.1.

## G   Supervised robust learning algorithms

We overview the algorithms of Montasser et al. [40, proofs of Theorems 4 and 8]. Their construction is based on sample compression methods explored in [32, 45].

Let $\mathcal{H} \subseteq \{0,1\}^{\mathcal{X}}$, fix a distribution $\mathcal{D}$ over the input space $\mathcal{X} \times \mathcal{Y}$. Let $S = \{(x_1, y_1), \ldots, (x_m, y_m)\}$ be an i.i.d. training sample from a robustly realizable distribution $\mathcal{D}$ by $\mathcal{H}$, namely $\inf_{h \in \mathcal{H}} \mathrm{Risk}_{\mathcal{U}}\left(h; \mathcal{D}\right) = 0$. Denote $d = \mathrm{VC}(\mathcal{H})$, $d^* = \mathrm{VC}^*(\mathcal{H})$ is the *dual VC-dimension*. Fix $\epsilon, \delta \in (0,1)$.

1. Define the inflated training data set

$$S_{\mathcal{U}} = \bigcup_{i \in [n]} \left\{ (z, y_{I(z)}) : z \in \mathcal{U}(x_i) \right\},$$

   where $I(z) = \min \{i \in [n] : z \in \mathcal{U}(x_i)\}$. The goal is to construct a compression scheme that is consistent with $S_{\mathcal{U}}$.

2. Discretize $S_{\mathcal{U}}$ to a finite set $\bar{S}_{\mathcal{U}}$. Define the class of hypotheses with zero robust error on every $d$ points in $S$,

$$\hat{\mathcal{H}} = \{\mathrm{RERM}_{\mathcal{H}}(S') : S' \subseteq S, |S'| = d\},$$

   where $\mathrm{RERM}_{\mathcal{H}}$ maps any labeled set to a hypothesis in $\mathcal{H}$ with zero robust loss on this set. The cardinality of this class is bounded as follows

$$|\hat{\mathcal{H}}| = \binom{n}{d} \leq \left( \frac{en}{d} \right)^d.$$

   Discretize $S_{\mathcal{U}}$ to a finite set using the finite class $\hat{\mathcal{H}}$. Define the *dual class* $\mathcal{H}^* \subseteq \{0,1\}^{\mathcal{H}}$ of $\mathcal{H}$ as the set of all functions $f_{(x,y)} : \mathcal{H} \to \{0,1\}$ defined by $f_{(x,y)}(h) = \mathbb{I}\left[h(x) \neq y\right]$, for any $h \in \mathcal{H}$ and $(x,y) \in S_{\mathcal{U}}$. If we think of a binary matrix where the rows consist of the distinct hypotheses and the columns are points, then the dual class corresponds to the transposed matrix where the distinct rows are points and the columns are hypotheses. A discretization $\bar{S}_{\mathcal{U}}$ will be defined by the dual-class of $\hat{\mathcal{H}}$. Formally, $\bar{S}_{\mathcal{U}} \subseteq S_{\mathcal{U}}$ consists of exactly one $(x,y) \in S_{\mathcal{U}}$ for each distinct classification $\left\{ f_{(x,y)}(h) \right\}_{h \in \hat{\mathcal{H}}}$. In other words, $\hat{\mathcal{H}}$ induces a finite partition of $S_{\mathcal{U}}$ into regions where every $\hat{h} \in \hat{\mathcal{H}}$ suffers a constant loss $\mathbb{I}\left[\hat{h}(x) \neq y\right]$ in each region, and the discretization $\bar{S}_{\mathcal{U}}$ takes one point per region. By Sauer's lemma [53, 47], for $n > 2d$,

$$|\bar{S}_{\mathcal{U}}| \leq \left( \frac{e|\hat{\mathcal{H}}|}{d^*} \right)^{d^*} \leq \left( \frac{e^2 n}{d d^*} \right)^{d d^*},$$

3. Execute the following modified version of the algorithm $\alpha$-boost [48, pages 162-163] on the discretized set $\bar{S}_{\mathcal{U}}$, with parameters $\alpha = \frac{1}{3}$ and number of boosting rounds $T = \mathcal{O}\left( \log\left(|\bar{S}_{\mathcal{U}}|\right) \right) = \mathcal{O}\left(d d^* \log(n)\right)$.

---

**Algorithm 2** Modified $\alpha$-boost

---

**Input:** $\mathcal{H}, S, \bar{S}_{\mathcal{U}}, d, \mathrm{RERM}_{\mathcal{H}}$.

**Parameters:** $\alpha, T$.

**Initialize** $P_1 = \mathrm{Uniform}(\bar{S}_{\mathcal{U}})$.

For $t = 1, \ldots, T$:

  (a) Find $\mathcal{O}(d)$ points $S_t \subseteq \bar{S}_{\mathcal{U}}$ such that every $h \in \mathcal{H}$ with $\widehat{\mathrm{R}}(h; S_t) = 0$ has $\mathrm{R}(h; P_t) \leq 1/3$.

  (b) Let $S'_t$ be the original $\mathcal{O}(d)$ points in $S$ with $S_t \subseteq \bigcup_{(x,y) \in S'_t} \bigcup \{(z,y) : z \in \mathcal{U}(x)\}$.

  (c) Let $\hat{h}_t = \mathrm{RERM}_{\mathcal{H}}(S'_t)$.

  (d) For each $(x, y) \in \bar{S}_{\mathcal{U}}$:
$$P_{t+1}(x, y) \propto P_t(x, y) e^{-\alpha \mathbb{I}\{\hat{h}_t(x) = y\}}$$

**Output:** classifiers $\hat{h}_1, \ldots, \hat{h}_T$ and sets $S'_1, \ldots, S'_T$.

---

4. Output the majority vote $\bar{h} = \mathrm{Majority}\left(\hat{h}_1, \ldots, \hat{h}_T\right)$.

We are guaranteed that $\widehat{\mathrm{R}}_{\mathcal{U}}\left(\bar{h}; S\right) = 0$, and each hypothesis $\hat{h}_t \in \left\{\hat{h}_1, \ldots, \hat{h}_T\right\}$ is representable as set $S'_t$ of size $\mathcal{O}(d)$. This defines a compression function $\kappa(S) = \bigcup_{t \in [T]} S'_t$. Thus, $\bar{h}$ can be reconstructed from a compression set of size

$$dT = \mathcal{O}\left(d^2 d^* \log(n)\right).$$

This compression size can be further reduced to $\mathcal{O}\left(dd^*\right)$, using a sparsification technique introduced by Moran and Yehudayoff [45], Hanneke et al. [32], by randomly choosing $\mathcal{O}(d^*)$ hypotheses from $\left\{\hat{h}_1, \ldots, \hat{h}_T\right\}$. The proof follows via a standard uniform convergence argument. Plugging it into a compression generalization bound, we have a sample complexity of $\tilde{\mathcal{O}}\left(\frac{dd^*}{\epsilon} + \frac{\log \frac{1}{\delta}}{\epsilon}\right)$, in the realizable robust case.

**Agnostic case.** The construction follows a reduction to the realizable case suggested by David et al. [24]. Denote $\Lambda_{\mathrm{RE}} = \Lambda_{\mathrm{RE}}(1/3, 1/3, \mathcal{H}, \mathcal{U})$, the sample complexity of $(1/3, 1/3)$-PAC learn $\mathcal{H}$ with respect to a perturbation function $\mathcal{U}$, in the realizable robust case.

Using a robust ERM, find the maximal subset $S' \subseteq S$ such that $\inf_{h \in \mathcal{H}} \widehat{\mathrm{R}}_{\mathcal{U}}\left(h; S'\right) = 0$. Now, $\Lambda_{\mathrm{RE}}$ samples suffice for weak robust learning for any distribution $\mathcal{D}$ on $S'$.

Execute the $\alpha$-boost algorithm [48, pages 162-163] on $S'$ for the robust loss function, with parameters $\alpha = \frac{1}{3}$ and number of boosting rounds $T = \mathcal{O}\left(\log\left(|S'|\right)\right)$, where each weak learner is trained on $\Lambda_{\mathrm{RE}}$ samples. The returned hypothesis $\bar{h} = \mathrm{Majority}\left(\hat{h}_1, \ldots, \hat{h}_T\right)$ satisfies that $\widehat{\mathrm{R}}_{\mathcal{U}}\left(\bar{h}; S'\right) = 0$, and each hypothesis $\hat{h}_t \in \left\{\hat{h}_1, \ldots, \hat{h}_T\right\}$ is representable as set of size $\mathcal{O}(\Lambda_{\mathrm{RE}})$. This defines a compression scheme of size $\Lambda_{\mathrm{RE}}T$, and $\bar{h}$ can be reconstructed from a compression set of points from $S$ of size $\Lambda_{\mathrm{RE}}T$.

Recall that $S' \subseteq S$ is a maximal subset such that $\inf_{h \in \mathcal{H}} \widehat{\mathrm{R}}_{\mathcal{U}}\left(h; S'\right) = 0$ which implies that $\widehat{\mathrm{R}}_{\mathcal{U}}\left(\bar{h}; S\right) \leq \inf_{h \in \mathcal{H}} \widehat{\mathrm{R}}_{\mathcal{U}}\left(h; S\right)$. Plugging it into a compression generalization bound (Lemma F.1 holds for the robust loss function as well), we have a sample complexity of $\tilde{\mathcal{O}}\left(\frac{\Lambda_{\mathrm{RE}}}{\epsilon^2} + \frac{\log \frac{1}{\delta}}{\epsilon^2}\right)$, which translates into $\tilde{\mathcal{O}}\left(\frac{dd^*}{\epsilon^2} + \frac{\log \frac{1}{\delta}}{\epsilon^2}\right)$, in the agnostic robust case.