# OpenReview forum: "A Characterization of Semi-Supervised Adversarially Robust PAC Learnability"
_NeurIPS.cc/2022/Conference — NeurIPS 2022 Accept_

### Official Review · Reviewer_czBH · 2022-06-23

**Rating:** 8
**Confidence:** 4
**Soundness:** 4 excellent
**Presentation:** 4 excellent
**Contribution:** 4 excellent

**Summary:**

The authors consider the problem of adversarial PAC learning in a semisupervised setting.

+ they show that having access to marginal distribution can provide provable benefits for this setting for some concept classes
+ they give a semi-supervised learner that uses labeled samples to learn an accurate but non-robust hypothesis (using partial concept learning).
+ They then show that one can use unlabeled data and label them with the learned hypothesis to come up with a larger labeled data from a distribution that is close to the actual labeled distribution. They then use the known results for adversarially robust pac learning on this data
+ their analysis shows provable benefits of unlabeled data for adversarial pac learning (for some hypothesis classes).
+ the analysis is extended to the agnostic setting

**Questions:**

N/A

**Strengths And Weaknesses:**

Strengths:

+ the main paper is written quite nicely
+ the results are interesting: they show provable benefits of unlabeled data for pac learning under adversarial perturbations
+ the idea of using the framework of partial concepts as well as labeling the unlabeled data is quite nice. Since this work builds on the existence of partial concept learners and robust pac learners, any future improvements will likely improve this result too.
+ Because the work build on the two building blocks (partial concepts and adversarial learning), the analysis is rather simple, clean, and intuitive. I like that the authors do not try to overcomplicate things, offering a good ground for future advancements.

Weaknesses:

+ The work of M Darnstädt, HU Simon and B Szörényi on "Unlabeled data does provably help" provides some provable benefits of semisupervsed learning in the non-adversarial setting. I think it would make sense to mention this result especially since the reader may think that in the non-adversarial distribution-free setting there is no benefit in knowing the marginal or having unlabeled data.
+ The work of H Ashtiani, V Pathak, R Urner  on "Black-box certification and learning under adversarial perturbations" provides some results on semi-supervised adversarial pac learning. It would help to discuss or compare the results.

---

> ### Author Response · Authors · 2022-07-31
> **Rebuttal**
>
> We thank the reviewer for the thoughtful and constructive feedback and for acknowledging that a simple algorithm and clean analysis are actually an advantage.
> We appreciate the suggestion about the related work section, see the comments below.
>
> $\textbf{Weaknesses}$:
>
> - "The work of M Darnstädt, HU Simon and B Szörényi on "Unlabeled data does provably help" provides some provable benefits of semi-supervised learning in the non-adversarial setting. I think it would make sense to mention this result especially since the reader may think that in the non-adversarial distribution-free setting there is no benefit in knowing the marginal or having unlabeled data".
>
> Answer: Thanks for pointing it out. We will definitely mention this paper in the next version. We will also make it clear that semi-supervised learning can be helpful for the non-adversarial case, but in a different sense than the distribution-free case.
>
> - "The work of H Ashtiani, V Pathak, R Urner on "Black-box certification and learning under adversarial perturbations" provides some results on semi-supervised adversarial pac learning. It would help to discuss or compare the results".
>
> Answer: Thanks for pointing it out. This is indeed very relevant, and we will dedicate a paragraph in the related work for discussing this paper in detail.

---

### Official Review · Reviewer_tNtE · 2022-07-11

**Rating:** 6
**Confidence:** 4
**Soundness:** 4 excellent
**Presentation:** 4 excellent
**Contribution:** 2 fair

**Summary:**

The authors tackle the learning of an adversarially robust predictor in the semi-supervised PAC model. They study the sample complexity for labeled and unlabeled examples for PAC learning. Their main focus is to control sample complexity of labeled examples which are more expensive to obtain than unlabeled examples. The sample complexity of unlabeled examples matches with that of labeled examples in supervised learning. Their main contribution is to bound sample complexity of labeled examples by a different complexity measure $VC_{\mathcal{U}}$. They provide new bounds for the following cases (I am mentioning only the labeled complexity):
1. Realizable and known support of the marginal distribution: $\Theta(\frac{VC_{\mathcal{U}}}{\epsilon} + \frac{log \frac{1}{\delta}}{\epsilon})$.
2. General realizable case: $\tilde{\mathcal{O}}(\frac{VC_{\mathcal{U}}}{\epsilon} + \frac{log \frac{1}{\delta}}{\epsilon})$.
3. General agnostic case: $\tilde{\mathcal{O}}(\frac{VC_{\mathcal{U}}}{\epsilon^2} + \frac{log \frac{1}{\delta}}{\epsilon^2})$ with error $2 \eta + \epsilon$ or $\tilde{\mathcal{O}}(\frac{RS_{\mathcal{U}}}{\epsilon^2} + \frac{log \frac{1}{\delta}}{\epsilon^2})$ with error $\eta + \epsilon$. They also show that $(\frac{3}{2} - \gamma)\eta + \epsilon$ error is unavoidable if the learner has access to only $\mathcal{O}(VC_{\mathcal{U}})$ labeled examples.

On a high-level the analysis can be summarized in the following steps:
1. Given the class $\mathcal{H}$, construct the partial concept class $\mathcal{H}_\mathcal{U}^*$.
2. Execute the learning algorithm for partial concepts to learn hypothesis $h_1$ for partial concept class $\mathcal{H}_\mathcal{U}^*$ . Bound sample complexity in terms of $\mathcal{O}(VC_\mathcal{U})$.
3. Label the unlabeled data set using $h_1$ .
4. Execute the agnostic adversarially robust supervised PAC learner to learn hypothesis $h_2$. Provide bounds on robust error.


**Questions:**

I liked the way the paper is presented and written. I will rephrase the weaknesses mentioned in previous section as questions, so that authors can respond to them.

1. Consider that results in [2] and [39] are well-known. Could you specify particular places (in concepts or theoretical analysis) where contribution is novel and is independent of contribution in [2] and [39]?
2. Could you provide some useful examples for specific $\mathcal{H}$ and $\mathcal{U}$ when $VC_\mathcal{U}$ is strictly smaller than $VC$ and the gain in complexity is significant?

**Limitations:**

The limitations have been adequately addressed.

**Strengths And Weaknesses:**

Strengths:
- The paper provides novel theoretical bounds for semi-supervised learning setting and is able to keep the labeled sample complexity down for the case when  $VC_\mathcal{U}$ is strictly smaller than $VC$.
- The paper is well written and theoretical results are presented in a manner which is easy to follow. In my opinion, the main contribution is in Definition 4.1 where they construct a partial concept class.

Weakness:
- The paper reads like an extension (although non-trivial) of ideas first mentioned in [2] and [39] to semi supervised learning setting. Once Definition 4.1 is established, the analysis directly borrows ideas from [2] and [39].
- The paper lacks some concrete examples for specific $\mathcal{H}$ and $\mathcal{U}$ when $VC_\mathcal{U}$ is strictly smaller than $VC$ and the gain in complexity is significant.

---

> ### Author Response · Authors · 2022-07-31
> **Rebuttal**
>
> We thank the reviewer for the thoughtful and constructive feedback.
>
> $\textbf{Questions}$:
>
> - Q: "Consider that results in [2] and [39] are well-known. Could you specify particular places (in concepts or theoretical analysis) where the contribution is novel and is independent of contribution in [2] and [39]?".
>
> Answer:
> As we reply to reviewer LSj4, we would like to clarify our view on the novelty.
> Our results have both conceptual aspects and technical contributions. The main conceptual aspect is to show that robust semi-supervised learning can, also, in theory, outperform robust supervised learning. This is the only distribution-free model (that we are aware of) with such a stark separation. This gap between the models is a curious property of robust learning. For example, in the standard semi-supervised PAC model, an analogous result does not hold. Moreover, we discovered an interesting and different structure for agnostic robust learning compared to the realizable case. We provide also a lower bound (Thm 5.3), showing that this gap is inherent to this learning model and not due to the presented algorithm.
>
> Our second contribution is indeed a skillful application of multiple methodologies in a surprising and highly non-trivial way. Incorporating partial concept classes in an adversarially robust setting is new (to our knowledge) and surprising. It allows us to provide a clear and elegant analysis of our algorithm. The fact that the resulting algorithm is simple and intuitive is yet another major benefit in our view.
>
> We would like also to mention the result in the section “Learning with the 0-1 loss assuming robust realizability”. This shows the benefit of using partial concept classes for learning with improved sample complexity, and, maybe surprisingly, we show that it cannot be obtained by a proper learning algorithm.
>
> - Q: "Could you provide some useful examples for specific H and U when VCU is strictly smaller than VC and the gain in complexity is significant?"
>
> Answer: Good point.
>  As one simple and natural example where VC_U << VC:
> H is linear separators on X the unit ball in R^d, and U as l_2 balls of radius gamma, the robustly realizable distributions are separable with margin gamma, where VC_U(H) = 1/gamma^2 but VC(H) = d+1 can be arbitrarily larger.

---

### Official Review · Reviewer_tkTj · 2022-07-25

**Rating:** 7
**Confidence:** 4
**Soundness:** 4 excellent
**Presentation:** 3 good
**Contribution:** 3 good

**Summary:**

The authors study semi-supervised learning for adversarially robust classification and claim upper bounds on labelled and unlabelled sample complexity of their learner. In particular, for the case when the data distribution is robustly realizable wrt (H, U) (i.e. when the robust loss is 0), they claim that their learner needs only a linear (in VC-dimension) number of labelled samples, as long as it's given a potentially exponential (in VC) number of unlabelled samples. More specifically, they show a linear dependence on a dimension that is defined in term of "robust shattering" with respect to the general perturbation sets U. It is known that this dimension can be significantly smaller than the VC-dimension. They also claim similar results for the agnostic case, and make additional remarks on special cases when their sample complexity can be improved further, and complement their upper bounds with corresponding lower bounds on the labeled and unlabeled samples.

**Questions:**

I think if we just want to show a linear dependence on VC-dimension for the number of labelled samples, we do not need to invoke the partial concepts paper?

**Limitations:**

The authors have addressed limitations adequately.

**Strengths And Weaknesses:**

The paper is sound and makes interesting new contributions.

---

> ### Author Response · Authors · 2022-07-31
> **Rebuttal**
>
> We thank the reviewer for the thoughtful feedback and for pointing out the observation below.
>
> $\textbf{Questions}$:
>
> - Q: "I think if we just want to show a linear dependence on VC-dimension for the number of labeled samples, we do not need to invoke the partial concepts paper?"
>
> Answer: Yes. This is a very nice observation! We think that you are right. In this case, roughly VC(H) labeled samples and RS(H) unlabeled samples are sufficient for learning.

---

### Official Review · Reviewer_LSj4 · 2022-07-26

**Rating:** 5
**Confidence:** 3
**Soundness:** 3 good
**Presentation:** 3 good
**Contribution:** 2 fair

**Summary:**

The paper considers realizable and agnostic semi-supervised PAC learning models. A common algorithm for robust semi-supervised learning is provided which first learns a partial concept class which predicts * (undefined) on points for which the concept may change prediction under an adversarial perturbation, using the labeled examples. This concept is used to extrapolate labels for the unlabeled examples on which an agnostic robust supervised learner is trained. Upper and lower bounds on the number of labeled examples needed are given in terms of the adversarial VC dimension, VC_U, in both realizable and agnostic settings. Further an application is provided, where the non-robust sample complexity is improved under a robust realizability assumption.

**Questions:**

Knowledge of supp(D_X) seems like a strong assumption. Are the authors aware of instances of this assumption in prior work?
Lines 253 vs 263: what does it mean that D_x is known? Is it the same as supp(D_x) is known?

What is 'labeled sampled complexity' (e.g. in Theorem 3.1)? Not defined in Sec 2. It seems like a bound on m_l in Definitions 2.2 and 2.4 but for what m_u (since definition is for pairs m_l,m_u)?

Similarly, in Theorem 5.2,5.3 what does m_u=\infty mean? Does it imply that m_l satisfies the lower bound for any (finite) m_u? Should be useful to add a remark in the context of Definitions 2.2 and 2.4.

In algorithm 1, h_1 is a partial concept class, right? How is the agnostic robust supervised learner applied to partially labeled sample S^u?

Typos:
155: exists
First para of Sec 2: definition of supp(D_X) should probably have an existential quantifier? \exists y\in Y s.t. D(x,y)>0?


**Limitations:**

I do not expect negative societal impact from this work.

**Strengths And Weaknesses:**

The paper shows a remarkable result for the number of labeled examples needed for robust semi-supervised learning - the results are somewhat surprising since finite VC_U is known to be not sufficient in the supervised setting (this is a strength).

Upper and lower bounds are provided in both realizable and agnostic settings. In the agnostic setting, the learner's goal is to have error const*OPT+epsilon since improvement over the supervised setting is not possible with OPT+epsilon.

Technically, the paper is an application of prior works on learning partial concept classes and agnostic robust supervised learning. The main novelty is combining the techniques.

It is not clear if one can say that VC_U is the correct sample complexity measure for the problem in all regimes of unlabeled sample size. It seems true in the sufficient unlabeled examples regime, so perhaps the stated results (e.g. in the abstract) should be qualified further.

Proof of Theorem 3.1 is not presented. The paragraph leading to it gives an explanation, but a formal proof should still ideally be included for completeness.

---

> ### Author Response · Authors · 2022-07-31
> **Rebuttal**
>
> We thank the reviewer for the thoughtful and constructive feedback.
>
> We would like to clarify our view on the novelty.
> Our results have both conceptual aspects and technical contributions. The main conceptual aspect is to show that robust semi-supervised learning can, also, in theory, outperform robust supervised learning. This is the only distribution-free model (that we are aware of)  with such a stark separation. This gap between the models is a curious property of robust learning. For example, in the standard semi-supervised PAC model, an analogous result does not hold. Moreover, we discovered an interesting and different structure for agnostic robust learning compared to the realizable case. We provide also a lower bound (Thm 5.3), showing that this gap is inherent to this learning model and not due to the presented algorithm.
>
> Our second contribution is indeed a skillful application of multiple methodologies in a surprising and highly non-trivial way. Incorporating partial concept classes in an adversarially robust setting is new (to our knowledge) and surprising. It allows us to provide a clear and elegant analysis of our algorithm. The fact that the resulting algorithm is simple and intuitive is yet another major benefit in our view.
>
> We would like also to mention the result in the section “Learning with the 0-1 loss assuming robust realizability”. This shows the benefit of using partial concept classes for learning with improved sample complexity, and, maybe surprisingly, we show that it cannot be obtained by a proper learning algorithm.
>
> $\textbf{Weaknesses}$:
>
> - "It is not clear if one can say that VC_U is the correct sample complexity measure for the problem in all regimes of unlabeled sample size. It seems true in the sufficient unlabeled examples regime, so perhaps the stated results (e.g. in the abstract) should be qualified further".
>
> Answer: Figure 1 demonstrates what is feasible/unfeasible labeled and unlabeled number of examples, but it is true that there is a gap as shown in figure 1.
> We will be very careful to state it correctly in the text.
>
> - "Proof of Theorem 3.1 is not presented. The paragraph leading to it gives an explanation, but a formal proof should still ideally be included for completeness".
>
> Answer: Thanks for the suggestion. For completeness. we will add a full explanation in the appendix.
> The proof is based on the following observation. Note that VC(H_(U-cons))=VC_U(H), given that, the bounds are straightforward from known results (see the paragraph above Thm 3.1).
>
> $\textbf{Questions}$:
>
> - Q: "Knowledge of supp(D_X) seems like a strong assumption. Are the authors aware of instances of this assumption in prior work? Lines 253 vs 263: what does it mean that D_X is known? Is it the same as supp(D_X) is known?"
>
> Answer:
> We agree that knowledge of supp(D_X) is a strong assumption but we use this simple case just for building the intuition for the general result. In Theorems 4.2 and 4.3 we prove the general case.
>
> In line 253, we write “support of the marginal distribution D_X is known” -  this is the same as supp(D_X) is known.
>
> In line 263, we write “when the marginal distribution D_X is known” - indeed this is stronger than knowing supp(D_X).
> However, our proof requires only knowing the supp(D_X) and we will change the Theorem statement accordingly.
>
> - Q: "What is 'labeled sampled complexity' (e.g. in Theorem 3.1)? Not defined in Sec 2. It seems like a bound on m_l in Definitions 2.2 and 2.4 but for what m_u (since the definition is for pairs m_l,m_u)?"
>
> Answer:
> We define the sample complexity by the pair m_l and m_u, where m_l is the “labeled sample complexity” and m_u is the “unlabeled sample complexity”.
> In theorem 3.1 (warm-up section)  we assume that supp(D_X) is known, and as a result, we do not require any unlabeled examples (m_u=0).
> We will add a remark to make it clear.
>
> - Q: "Similarly, in Theorem 5.2,5.3 what does m_u=\infty mean? Does it imply that m_l satisfies the lower bound for any (finite) m_u? Should be useful to add a remark in the context of Definitions 2.2 and 2.4".
>
> Answer: Yes.
> It means that as long as the labeled sample size satisfies the lower bound, any finite unlabeled sample size is not sufficient in order to ensure learning.
> We will add a remark to make it clear.
>
> - Q: "In algorithm 1, h_1 is a partial concept class, right? How is the agnostic robust supervised learner applied to partially labeled sample S^u?"
>
> Answer: Good point. In the agnostic robust case, h_1 might return *. We can label these points however we want, and it will not affect the correctness of the algorithm.
> Thank you for pointing it out. We will make it clear.
> (side note: In the robust realizable case, h_1 will not return *)
>
> - Q: "Typos: 155: exists First para of Sec 2: definition of supp(D_X) should probably have an existential quantifier? \exists y\in Y s.t. D(x,y)>0?"
>
> Answer: Thanks! it will be corrected.

---

### Meta-Review · Area_Chair_mR8p · 2022-08-25

**Recommendation:** Accept
**Confidence:** Certain

**Metareview:**

This submission provides novel results on the benefits of unlabelled data (through a semi-supervised learning framework) for adversarial robust PAC learning. The results are correct, novel and will be of substantial interest to the corresponding sub-community of NeurIPS. I therefore recommend acceptance.


Side note: the following publication studies SSL (PAC-type) sample complexity (upper and lower) bounds with the same algorithmic idea, and I believe should be acknowledged:

Ruth Urner, Shai Shalev-Shwartz, Shai Ben-David: Access to Unlabeled Data can Speed up Prediction Time. ICML 2011: 641-648

**Award:**

No

---

### Decision · Program_Chairs · 2022-09-14

Accept